# Unbiased identification of novel transcription factors in striatal compartmentation and striosome maturation

Maria-Daniela Cirnaru[1], Sicheng Song[2], Kizito-Tshitoko Tshilenge[3], Chuhyon Corwin[1], Justyna Mleczko[1], Carlos Galicia Aguirre[3], Houda Benlhabib[2], Jaroslav Bendl[4,5,6], Pasha Apontes[4,5,6], John Fullard[4,5,6], Jordi Creus-Muncunill[1], Azadeh Reyahi[7], Ali M Nik[7], Peter Carlsson[7], Panos Roussos[4,5,6,8], Sean D Mooney[2], Lisa M Ellerby[3], Michelle E Ehrlich[1]*

[1]Department of Neurology, Icahn School of Medicine at Mount Sinai, New York, United States; [2]Department of Biomedical Informatics and Medical Education, University of Washington, Seattle, United States; [3]Buck Institute for Research on Aging, Novato, United States; [4]Pamela Sklar Division of Psychiatric Genomics, Icahn School of Medicine at Mount Sinai, New York, United States; [5]Institute for Genomics and Multiscale Biology, Department of Genetics and Genomic Sciences, Icahn School of Medicine at Mount Sinai, New York, United States; [6]Department of Psychiatry, Icahn School of Medicine at Mount Sinai, New York, United States; [7]Department of Chemistry and Molecular Biology, University of Gothenburg, Gothenburg, Sweden; [8]Mental Illness Research, Education, and Clinical Center (VISN 2 South), Bronx, United States

*For correspondence:
michelle.ehrlich@mssm.edu

Competing interests: The authors declare that no competing interests exist.

**Abstract** Many diseases are linked to dysregulation of the striatum. Striatal function depends on neuronal compartmentation into striosomes and matrix. Striatal projection neurons are GABAergic medium spiny neurons (MSNs), subtyped by selective expression of receptors, neuropeptides, and other gene families. Neurogenesis of the striosome and matrix occurs in separate waves, but the factors regulating compartmentation and neuronal differentiation are largely unidentified. We performed RNA- and ATAC-seq on sorted striosome and matrix cells at postnatal day 3, using the *Nr4a1*-EGFP striosome reporter mouse. Focusing on the striosome, we validated the localization and/or role of *Irx1*, *Foxf2*, *Olig2*, and *Stat1/2* in the developing striosome and the in vivo enhancer function of a striosome-specific open chromatin region 4.4 Kb downstream of *Olig2*. These data provide novel tools to dissect and manipulate the networks regulating MSN compartmentation and differentiation, including in human iPSC-derived striatal neurons for disease modeling and drug discovery.

## Introduction

Dysregulation of the striatum is linked to multiple neuropsychiatric diseases, including Huntington's (HD), Parkinson's, X-linked dystonia-parkinsonism, addiction, autism, and schizophrenia. The dorsal striatum comprises the caudate and putamen in humans but consists of a single nucleus in the mouse. This nucleus is a key component of cortical and subcortical circuits regulating movement, reward, and aspects of cognition, including speech and language. 85 to 95% of striatal neurons are medium spiny (MSNs), the striatal projection neuron. They are morphologically homogeneous, but

phenotypically heterogeneous, and adult subtypes are distinguished by unique transcriptomes (*Anderson et al., 2020*; *Gokce et al., 2016*; *Ho et al., 2018*; *Märtin et al., 2019*; *Muñoz-Manchado et al., 2018*; *Ortiz et al., 2020*; *Stanley et al., 2020*; *Zeisel et al., 2018*). MSNs are equally distributed between direct neurons (dMSNs), which express the dopamine D1 receptor (D1R) and project directly to the substantia nigra (SN) or to the internal segment of the globus pallidus, and indirect neurons (iMSNs), which express the dopamine D2 (D2R) and adenosine 2A receptors (A2aR) and project to the external segment of the globus pallidus or to the subthalamic nucleus (*Keeler et al., 2014*).

The striatum is comprised of two main compartments named striosomes (patch) and matrix. Striosomes occupy 10–15% of the volume and are dispersed in a continuum throughout the 80–85% occupied by the matrix. Importantly, an imbalance between striatal compartments likely contributes to movement disorders (*Cazorla et al., 2015*; *Crittenden and Graybiel, 2011*; *Keeler et al., 2014*), and compartmentation also appears to be required for non-motor functions, for example, speech and language and discrimination learning (*Campbell et al., 2009*; *Deriziotis and Fisher, 2017*; *Feuk et al., 2006*; *Konopka et al., 2009*; *Lai et al., 2001*; *MacDermot et al., 2005*). In X-linked dystonia-parkinsonism, degeneration is initiated in the striosomes (*Beste et al., 2018*; *Goto et al., 2005*). Compartmentation is altered in autism (*Kuo and Liu, 2020*), and HD features early, preferential loss of striosome neurons (*Victor et al., 2014*). Moreover, dopaminergic signaling has opposing effects on D1R activation in a compartment-specific manner, which is important for task dependent behaviors (*Prager et al., 2020*). The distinction between the striosome and matrix in adults is based on differences in gene expression, the origins of afferents from cortical regions (e.g. sensorimotor, limbic, and associative), and to some extent, by the destination of their efferents (*Brimblecombe and Cragg, 2017*; *Crittenden and Graybiel, 2011*; *Fujiyama et al., 2019*). They both contain direct and indirect neurons, but the striosome dMSN content regionally varies (*Cirnaru et al., 2019*; *Miyamoto et al., 2018*). Currently, no robust differentiation protocol is available for the generation of striosome cells for in vitro studies and replacement therapies (*Arber et al., 2015*; *Golas, 2018*; *Kemp et al., 2016*; *Richner et al., 2015*; *Telezhkin et al., 2016*; *Victor et al., 2014*), and the molecular mechanisms governing compartmentation are incompletely defined.

Striosome and matrix compartments differ in their developmental timelines, as there are two waves of striatal neurogenesis (*Matsushima and Graybiel, 2020*). Striosome MSNs complete mitosis by embryonic day 13 (E13) and project toward substantia nigra pars compacta by E18 (*Fishell and van der Kooy, 1987*; *Fishell and van der Kooy, 1989*). Matrix MSNs are born in the second wave and are postmitotic by E18–E20 and do not project toward substantia nigra until the first postnatal week (*Fishell and van der Kooy, 1987*; *Fishell and van der Kooy, 1989*). This period is essential for the specification of striatal compartmentation, and most of the markers that distinguish striosome from matrix do not reach their final adult distribution until at least the second postnatal week. For example, dopamine- and cAMP-regulated neuronal phosphoprotein (DARPP-32), encoded by *Ppp1r1b*, is expressed in 95% of the MSNs at maturity and is the most commonly used pan-MSN marker (*Arber et al., 2015*; *Chandwani et al., 2013*; *Fjodorova et al., 2019*; *Fullard et al., 2018*; *Ivkovic and Ehrlich, 1999*), but it is unequivocally a marker of striosomes during the first postnatal week (*Arlotta et al., 2008*). Mu opiate receptor (MOR) protein, encoded by the *Oprm1* gene, marks the striosome neurons at all ages but conversely, calbindin, encoded by *Calb1*, is the most widely used matrix marker in studies of adult striatum, but at PND3, is expressed in nascent striosomes (*Anderson et al., 1997*; *Arlotta et al., 2008*; *Crittenden and Graybiel, 2011*; *Liu and Graybiel, 1992*; *Passante et al., 2008*; *Snyder-Keller et al., 2008*). A recent study analyzing P9 striosome and matrix confirms that maturation is not yet complete, based on comparisons to distribution of adult markers (*Anderson et al., 2020*). The developmental cascade for the two compartments is shown schematically in the upper panel of *Figure 1*.

Multiple transcription factors (TFs) are required for induction and differentiation into MSN subtypes (*Arlotta et al., 2008*; *Fjodorova et al., 2019*; *Golas, 2018*; *Ivkovic and Ehrlich, 1999*; *Long et al., 2009*; *Marin et al., 2000*; *Martín-Ibáñez et al., 2012*; *Martín-Ibáñez et al., 2017*; *Precious et al., 2016*; *Victor et al., 2014*; *Wang et al., 2011b*; *Zhang et al., 2016*), and some are useful for direct conversion of human fibroblasts into MSNs (*Hedreen and Folstein, 1995*; *Lawhorn et al., 2008*). Importantly, a widely used protocol yielded only a matrix phenotype (*Victor et al., 2014*), and several others yielded largely calbindin-positive neurons, with no mention

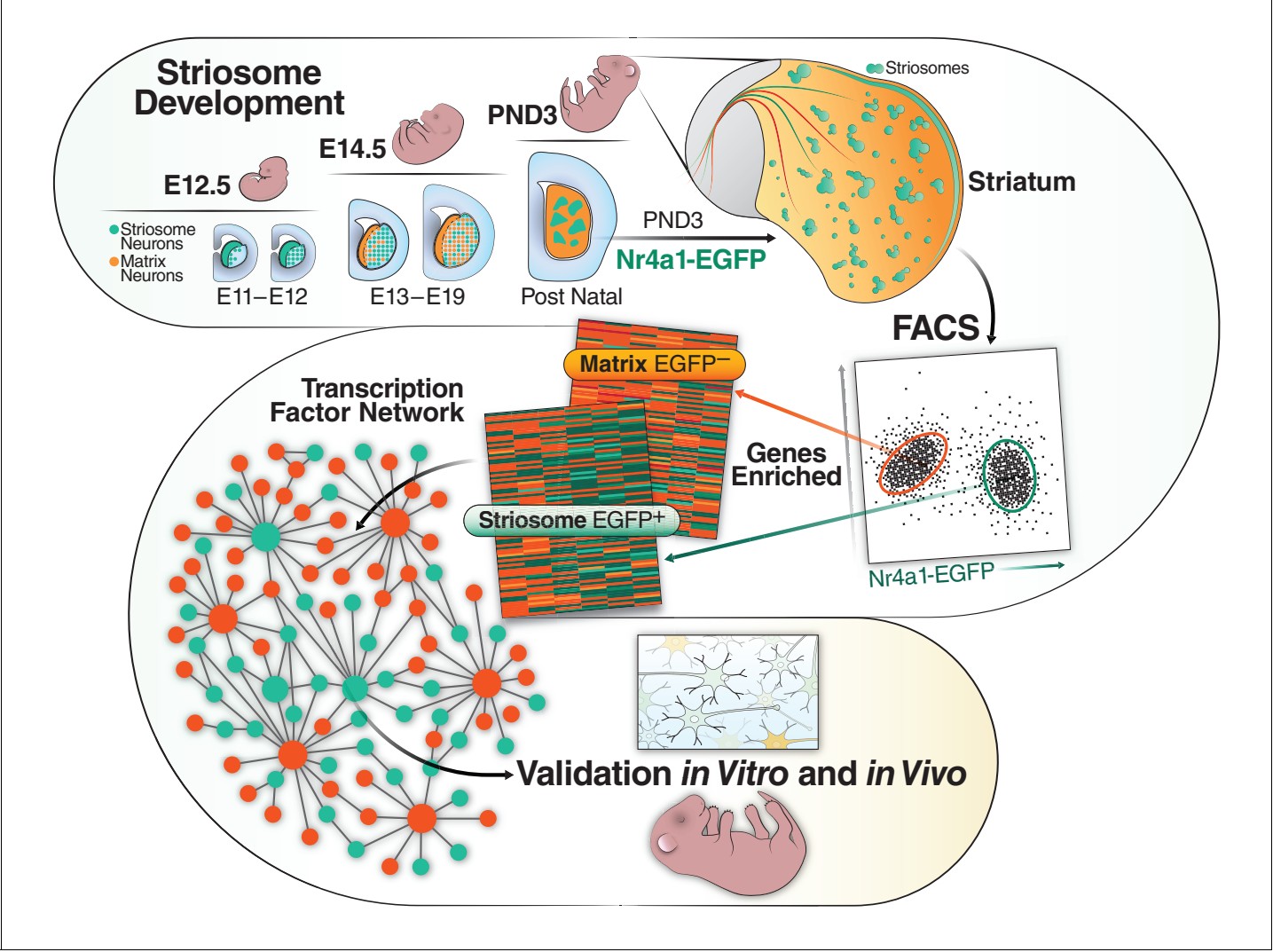

**Figure 1.** Schematic of the developmental cascade for the striosome and matrix compartments and our experimental approach. Upper panel shows the cascade of the two compartments in the developing mouse. Lower panels show our experimental approach. We FACs sorted EGFP⁺ cells from Nr4a1 EGFP mice at PND3, performed RNAseq, identified top transcription factors in the striosome, validated them in vivo (mice) and in vitro (primary mouse cultures and human iPSCs).

of a striosome phenotype (*Adil et al., 2018*; *Arber et al., 2015*; *Golas, 2018*; *Kemp et al., 2016*; *Telezhkin et al., 2016*). Only a handful of TFs and signaling systems contributing to striosome/ matrix compartmentation have been identified, including *Ikaros-1*, retinoids, *Foxp1*, and *Nr4a1* (*Cirnaru et al., 2019*; *Evans et al., 2012*; *Kuo and Liu, 2017*; *Lai et al., 2001*; *Martín-Ibáñez et al., 2012*; *Martín-Ibáñez et al., 2017*; *Rataj-Baniowska et al., 2015*), the absence of which in some cases reduces the area occupied by striosomes but does not prevent the formation of the basic architecture.

We used unbiased transcriptomic and epigenetic assays, RNA-seq and ATAC-seq, to identify TFs and open chromatin regions (OCRs) associated with striosome maturation on postnatal day 3, specifically after they are first formed but not yet mature (*Figure 1*). Genes expressed at this point may also include terminal differentiation effectors as defined by *Hobert, 2016*, which initiate and maintain the adult identity of neurons. We used the GENSAT *Nr4a1*-EGFP mouse, in which reporter expression is directed to striosomes throughout the life of the mouse (*Cirnaru et al., 2019*; *Davis and Puhl, 2011*), including as specifically described on PND3. As we did not know which of the adult striosome and matrix markers were already expressed in their respective, final compartments on PND3 which could be used to assign neuronal identity, we opted to perform bulk RNA-

seq on cells sorted by fluorescence-activated cell sorting (FACS) rather than single cells, also possibly increasing the capture of low expression genes (*Figure 1*). We developed complementary databases of the EGFP-negative cells, enriched in developing matrix neurons. We validated the localization and/or role of *Irx1*, *Foxf2*, *Olig2*, and *Stat1/2* in the developing mouse striosome and the in vivo enhancer function of a striosome-specific open chromatin region 4.4 Kb downstream of the 3' end of the *Olig2* gene. We also show that *Foxf2*, *Olig2*, and *Stat1/2* drive the expression of MSN maturation in human NSCs derived from HD-iPSCs. The identified TFs and OCRs greatly add to the understanding of MSN development and allow for the generation of additional MSN subtypes for disease modeling.

## Results

### Transcriptional analysis of *Nr4a1*-EGFP mice identify striatal compartment-specific TFs in vivo

The TF *Nr4a1* is a member of the Nur family of steroid/thyroid-like receptors (*Giguère, 1999*) and is expressed in the mouse striosome MSNs as early as embryonic day (E) 14.5 (*Davis and Puhl, 2011*). GENSAT mice express the EGFP reporter from a transgenic *Nr4a1* bacterial artificial chromosome (*Nr4a1*-EGFP), and at PND3, EGFP co-localizes with Oprm1, D1R, and tyrosine hydroxylase islands. We confirmed co-localization with Oprm1/MOR and further demonstrated that EGFP co-localizes to a large extent with DARPP-32 (*Figure 2—figure supplement 1A*). A list of striosome and matrix markers used in this report is presented in *Table 1*. Notably, the overlap with several of these markers is not 100% but does include the subcallosal streak. *Nr4a1*-EGFP-positive and -negative cell populations, representing enriched striosome and matrix MSN populations, respectively (*Figure 2B*), were isolated by FACS using spontaneous fluorescence. Visual inspection confirmed separation of the two populations (*Figure 2C*). RT-qPCR analysis of the sorted populations indicated significant enrichment of PND3 striosome markers *Ppp1r1b* and *Oprm1*, *Nr4a1* and GFP in the EGFP+ population (*Figure 2—figure supplement 1B*). *Map2*, associated with more mature neurons, was enriched, whereas glial fibrillary acidic protein (*Gfap*) mRNA was relatively lower in the EGFP+ sample, indicating neuronal enrichment in this fraction. *Tuj1*, associated with immature neurons (*Menezes and Luskin, 1994*) was enriched in the EGFP– population (*Figure 2—figure supplement 1B*). We concluded that the sorting captured a large subset of PND3 striosome neurons. The extent of inclusion of other cell types, including exopatch neurons, is discussed below.

Next, we carried out RNA-seq, transcriptomic analysis on both the EGFP+ and EGFP– cells (*Figure 2—source data 1*). A total of 9124 genes were differentially expressed between the two groups, with 4714 enriched in the EGFP+ cells (positive log2FC) and 4410 enriched in the EGFP– cells (negative log2FC) ($p<0.01$, *Figure 2—source data 1*), but this entire list includes many genes with small differences between compartments. Twenty-three differentially expressed genes were selected with

**Table 1.** List of striosome and matrix markers.

| Gene symbol | Gene name | PND3 | Adult |
|---|---|---|---|
| *Calb1* | Calbindin | Striosome | Matrix |
| *EfnA5* | Ephrin A5 | Striosome | |
| *EphA4* | Eph Receptor A4 | Matrix | Matrix |
| *EphA7* | Eph Receptor A7 | Striosome | |
| *FoxP1* | Forkhead Box P1 | Striosome | pan-MSN |
| *FoxP2* | Forkhead Box P2 | Striosome | Striosome |
| *Nr4a1* | Nuclear Receptor Subfamily 4 Group Member 1 | Striosome | Striosome |
| *Oprm1* | Mu opioid receptor | Striosome | Striosome |
| *Ppp1r1b* | Protein phosphatase 1 regulatory subunit 1B | Striosome | pan-MSN |
| *Rasgrp1* | CalDAG GEFII | Striosome | Striosome |
| *Rasgrp2* | CalDAG GEFI | Matrix | Matrix |

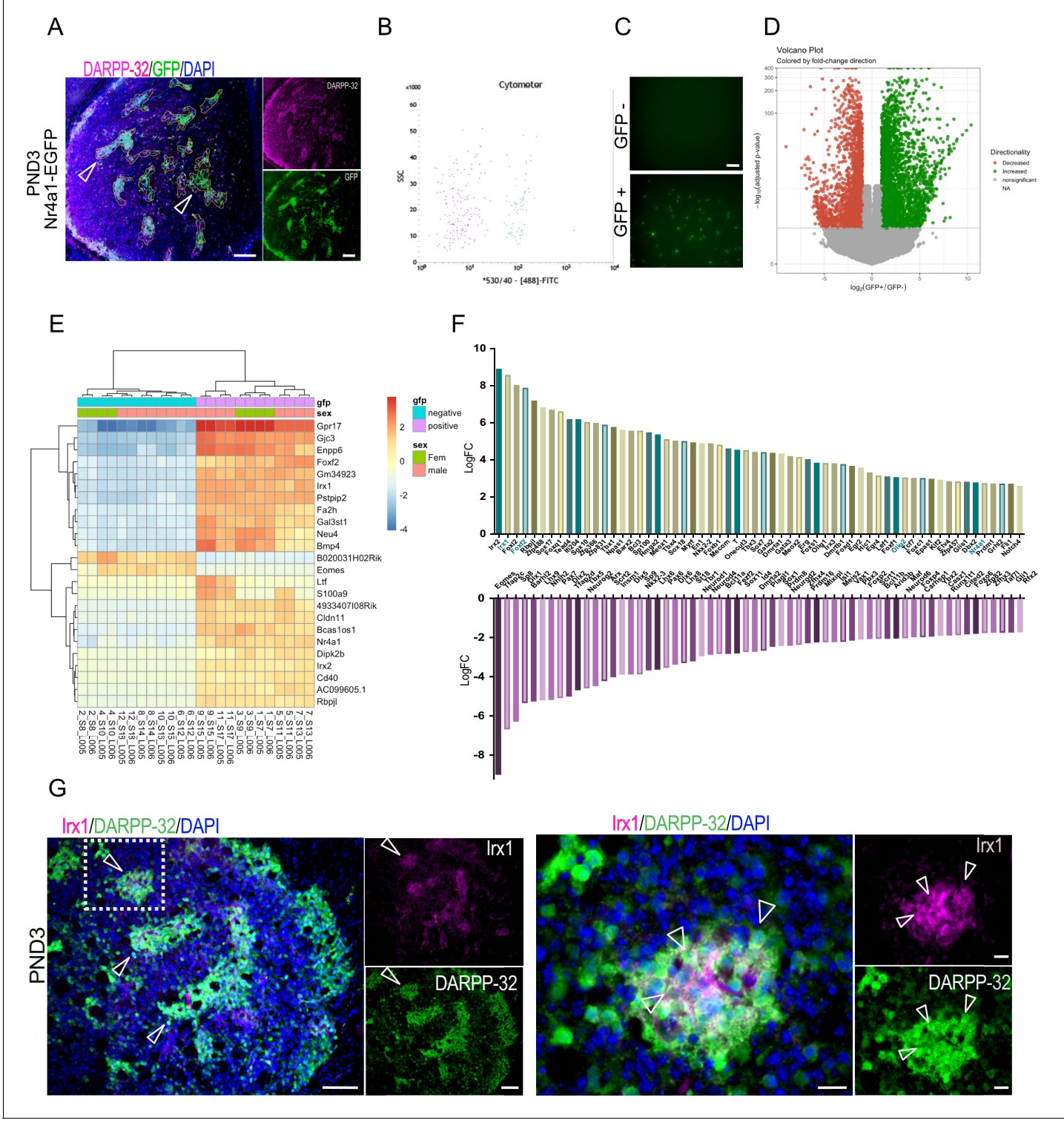

**Figure 2.** Identification of striatal striosome and matrix compartment-specific transcription factors using RNA-seq analysis in FACs EGFP$^+$ and EGFP$^-$ cell populations from PND3 *Nr4a1*- EGFP mice. (**A**) Coronal section of PND3 *Nr4a1*-EGFP mouse showing the striosomes (open arrowheads pointing to several examples) localization of spontaneous EGFP fluorescence, with DARPP-32 (magenta) colocalization. Scale bar = 100 µm. (**B**) Representative FACs cytometer panel indicating cell separation using the intensity of the 488-FITC channel with clear distribution of the EGFP$^+$ population on the right side of the panel. (**C**) Detection of spontaneous EGFP fluorescence of EGFP$^+$ and EGFP$^-$ cells seeded at 1000 cells/well 2 hr after FACs sorting, indicating the presence of EGFP exclusively in EGFP$^+$ samples. Scale bar = 50 µm. (**D**) Volcano plot visualizing the differences in gene expression between the EGFP$^+$ and EGFP$^-$ cell populations. Genes with an adjusted p-value below 0.01 with absolute log2 fold ratio greater than one are

*Figure 2 continued on next page*

*Figure 2 continued*

highlighted. Red indicates genes that were relatively decreased in expression in EGFP⁺ population (i.e. enriched in the EGFP⁻ population), green indicates genes that were relatively enriched in expression in the EGFP⁺ population, and gray indicates those genes that were equally distributed in the two populations. (E) Heatmap of relative normalized counts across samples of genes with an absolute log2 fold-change > 7, and mean normalized counts > 40. (F) Representation of the 60 TFs most enriched in either population, according to their log2 fold change in the EGFP⁺ population (upper panel) and in the EGFP⁻ population (lower panel), without regard to mean normalized counts. (G) Representative low- and high-power micrographs of Irx1 (magenta), DARPP-32 (green) and DAPI (blue) immunolabeling on 16-µm-thick coronal sections from wild-type PND3 showing the localization of Irx1 within a subset of the DARPP-32 immunopositive striosomes (arrowheads left panel), with double label of a subset of striosome neurons indicated by the arrowheads (right panel). Scale bars = 100 µm and 20 µm, respectively.

The online version of this article includes the following source data and figure supplement(s) for figure 2:

**Source data 1.** RNA sequencing analysis of the transcripomes of striosome and matrix cells from the PND3 striatum striosome and matrix compartments expressed as a ratio of Nr4a1-EGFP+/Nr4a1-EGFP- following FACS sorting.

**Figure supplement 1.** Validation of *Nr4a1*-EGFP striosome localization and characterization of EGFP+ and EGFP- cell population after FACS sorting.

an arbitrary threshold of log2FC greater than seven and mean normalized counts greater than 40, which along with *Nr4a1*, are shown in the heatmap (*Figure 2E*). The heatmap includes *Irx1, Irx2* and *Foxf2* as top candidates for a role in defining the striosome, based on their relative distribution.

The *Nr4a1* log2FC indicates highly significant [p(adj) = 1.47E-113] enrichment in the striosome compartment but suggests by its relatively lower value, +2.75, that the EGFP expression directed by the BAC transgene may not perfectly replicate endogenous *Nr4a1* expression.

To further highlight compartmentalized differentially expressed TFs, we used the mouse TF database (*Figure 2—source data 1*). The top 60 TFs enriched in each population according to their log2FC [i.e., EGFP+ population relative to the EGFP– (upper panel), and in the EGFP– population (lower panel), without regard to mean normalized counts] are shown in *Figure 2F*. TFs enriched in the EGFP⁺ group include *Foxf2* as noted above and *Olig2,* which are enriched seven- and threefold over their expression levels in the matrix, respectively. Gene sets that are enriched in the EGFP⁻ compartment include numerous known and novel genes that may have a role in the overall specification of the MSN and particularly matrix MSNs (*Arlotta et al., 2008*; *Fjodorova et al., 2019*; *Golas, 2018*; *Ivkovic and Ehrlich, 1999*; *Long et al., 2009*; *Marin et al., 2000*; *Martín-Ibáñez et al., 2012*; *Martín-Ibáñez et al., 2017*; *Precious et al., 2016*; *Victor et al., 2014*; *Wang et al., 2011b*; *Zhang et al., 2016*). Known striatal TFs include *FoxP1, Dlx1,2,6, Gsx2, Mash1, Nkx2.1, Lhx6,* and *Lhx7.* The latter three are associated with interneurons. The full list of TFs that define the striosome and matrix at this age are summarized in *Figure 2—source data 1*. We identified 621 differentially expressed TFs. Among these, 259 were enriched in the EGFP+ cells, and 362 were enriched in the EGFP– cells.

The ongoing maturation of the two compartments was further highlighted by the localization of other mRNAs recently associated with striosome, matrix, and exopatch (*Ortiz et al., 2020*; *Saunders et al., 2018*; *Smith et al., 2016*). The fact that some of these are not yet compartmentalized and/or not present in the data base confirms that both neuronal maturation and compartmentation are incomplete on PND3. Thus, adult and/or PND9 striosome markers *Sema5b, Kremen1, Tac1, Spon1, Sorcs1, Cdh18, Cdh10,* and *Asic4* are all enriched in the EGFP+ compartment, but *Mfge8, Sepw1,* and *LyPd1* are evenly distributed between EGFP+ and EGFP– cells, and *Pde1c* and *Tshz1* mRNAs are enriched in the EGFP– compartment, as is *Foxp2*. Of the exopatch markers *Casz1, Otof, Ntng1,* and *Asic2*, only the latter is enriched in the EGFP+ compartment, so it is unknown whether we are capturing exopatch neurons.

As the *Irx1/2* genes were among the most enriched TFs in the striosomes and have not been associated with the striatum, we carried out immunostaining of Irx1 to validate the predicted protein distribution. Irx1 co-localizes with a subset of DARPP-32 immunopositive striosomes in wild-type PND3 mice, validating its predicted striosome enrichment (*Figure 2G*).

## Analysis of striosome-enriched transcription networks define *Foxf2* and *Olig2* as key differentiation factors

To identify potential master regulators driving the transcription program of the striosome, we conducted TF and co-expressor enrichment analysis with Enrichr (*Chen et al., 2013*; *Kuleshov et al., 2016*). We input the list of either differentially expressed genes or differentially expressed

transcription factors to obtain the upstream regulators and co-expressors for both gene sets (p<0.05). The intersect of the enrichment results and the differentially expressed TFs were further used as input to generate striosome- and matrix-specific TF co-expression networks, using GeneMA-NIA (*Warde-Farley et al., 2010*) as a prediction tool (*Figure 3A,B*). This allowed the generation of network reconstruction and expansion for striosome and matrix differentially expressed TFs. From the network analysis, we observed that striosome-enriched *Nr4a1*, *Irx1*, *Olig2*, and *Foxf2* interacted with a large number of TFs (*Figure 3A*), indicating that these four TFs are likely critical in the development of striosomes and the MSNs located therein, as we already showed for *Nr4a1* (*Cirnaru et al., 2019*). Since *Foxf2* and *Olig2* are potential novel regulators of the striosome (*Figure 3E,F*), we asked whether or not their targets were co-regulated in this compartment. We found that, in the striosome, the transcriptional targets of *Foxf2* and *Olig2* are indeed upregulated, i.e., enriched, relative to the matrix, and predicted to be coordinately regulated (*Figure 3C,D*). The TF network analysis of the matrix (*Figure 3B*) indicates *Dlx1* and *Dlx2* are hub TFs. Potentially newly defined hub TFs in the matrix include *Pax7*, *Barhl2*, *Lhx9*, *Nhlh2*, *Eomes*, and *Sp8* (*Figure 3B*), the latter already associated with development of iMSNs (*Xu et al., 2018*).

In addition to defining the transcriptional network for the striosome and matrix, we analyzed the differentially expressed genes for overrepresented biological annotations or pathways, including GO term and KEGG pathway enrichment analysis (*The Gene Ontology Consortium, 2017*; *Figure 2— source data 1*). Striosome enriched genes were associated with general development, for example, integrins, angiogenesis or cell motility, and nodes include 'regulation of pre-synapse organization'. Matrix-enriched genes were associated with neural system development and neurotransmission (*Figure 3E*). Both *Nr4a1* and *Foxf2* are associated with angiogenesis and/or establishment of the blood brain barrier (*Reyahi et al., 2015*; *Zeng et al., 2006*), which play synergistic roles with brain and neuronal development (*Haddad-Tóvolli et al., 2017*). Calcium regulation was associated with both gene sets. We also conducted a similar analysis with the differentially expressed TFs. For visualization and interaction network and GO enrichment analysis, TFs in the EGFP+ compartment are highly involved in embryonic organ development (*Sox17*, *Gata3*, *Cebp*, *Nr4a3*, *Foxf1*, *Foxf2*), cell-fate commitment (*Sox2*, *Cebp*, *Stat3*, *Olig1* and *Olig2*) and transcription activation, including RNA polymerase II binding factors and enhancer binding (*Id1*, *Klf4*, *Tbx3*, *Erg2*, *Stat3*), indicating an active, open chromatin structure (*Figure 2—source data 1*).

## ATAC-seq analysis defines compartment-specific OCRs in the striatum

To map chromatin accessibility of striosome and matrix cells, we used FACS, followed by preparation of eight ATAC-seq libraries from 12 animals pooled in four independent samples for the EGFP+ and EGFP− populations. Overall, we obtained 322 million (average of 40.3 million) uniquely mapped reads after removing duplicates and those aligning to the mitochondrial genome (*Figure 4—source data 1*). To quantitatively analyze differences between striosome and matrix cells, we generated a consensus set of 69,220 OCRs by taking the union of peaks called in the individual cells (Materials and methods). We next quantified the number of reads that overlapped each OCR. Uniform Manifold Approximation and Projection for Dimension Reduction (UMAP)-based clustering using the normalized read counts clearly separated striosome from matrix samples (*Figure 4A*). Comparison of striosome and matrix peaks resulted in 44% of OCRs that were significant after multiple testing corrections (30,799 differentially modified OCRs at FDR $\leq$ 0.05) (*Figure 4—source data 2*). Among these, 16,963 were striosome, and 13,836 were matrix.

We examined the location of OCRs with respect to the distance from transcription start sites (TSS) and genic annotations. As expected, OCRs are in the vicinity of TSSs (*Figure 4B*) but are also enriched for non-promoter regulatory elements, suggesting a more important role for long-range regulation of gene expression in developing striosome and matrix. We tested the concordance of striosome and matrix-specific genes and OCRs, based on the RNA-seq and ATAC- seq analyses, respectively, and found a moderately high correlation (Pearson's R = 0.34; p-value < 2.2x10-16) (*Figure 4C*). To determine if the ATAC-seq identified regulatory regions relevant to *Nr4a1*-EGFP enriched expression in the striosome, we compared the tracks from PND3 EGFP+ and EGFP− populations for the presence of putative *Nr4a1* cis-regulatory elements (*Figure 4D*). Indeed, a number of distinct OCRs within the *Nr4a1* gene were associated with the EGFP+ cells. Next, we used HOMER (*Heinz et al., 2010*) to identify which TF binding motifs were selectively enriched in the OCRs of EGFP+ and EGFP− populations. De novo motif enrichment analysis of the EGFP+ OCRs showed

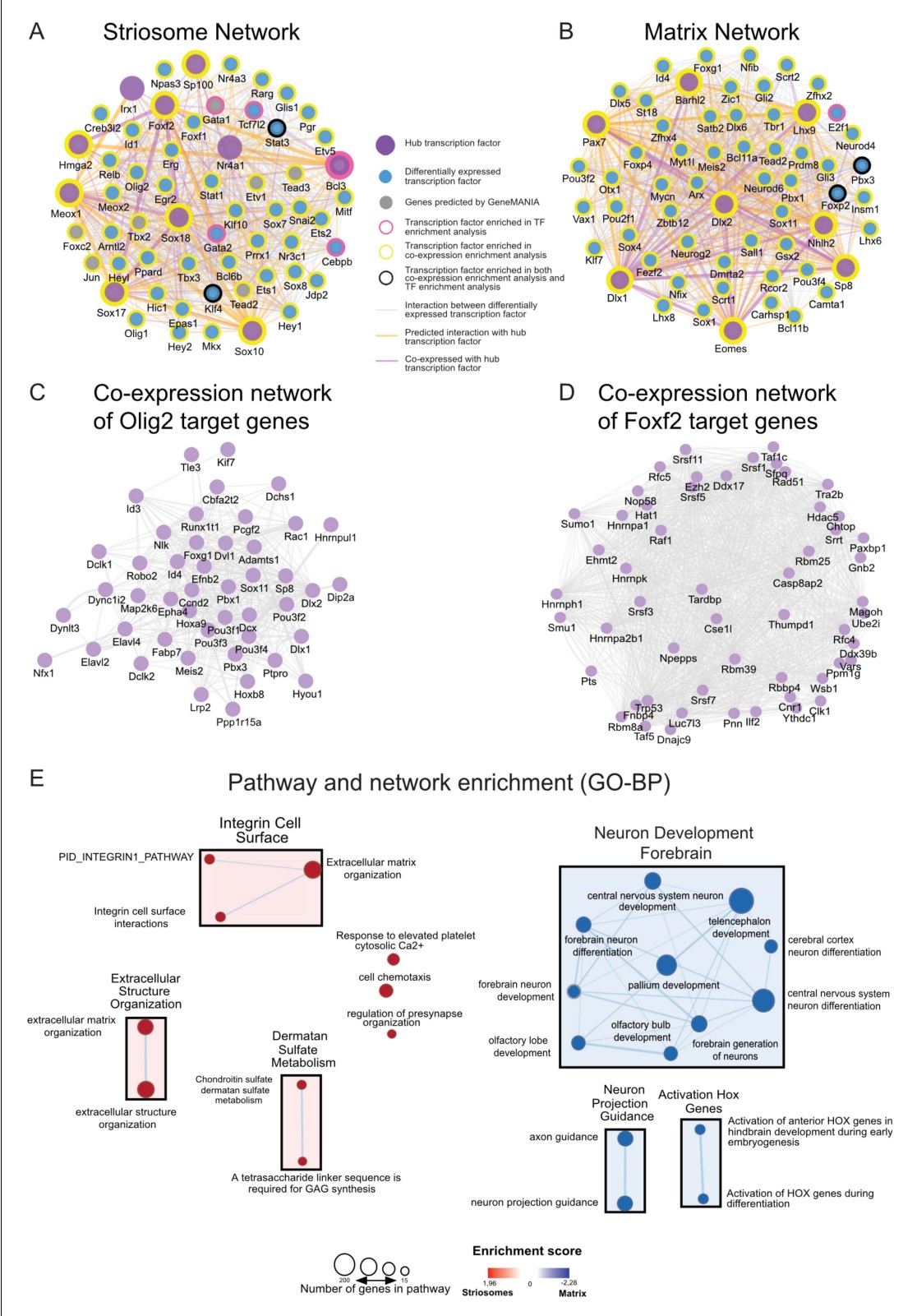

**Figure 3.** Transcription networks and enrichment maps of striosome and matrix. (**A,B**) TF networks enriched in either striosome or matrix show interactions and co-expression among the regulatory genes. An arbitrary threshold of absolute log2 fold-change > five was set to designate hub genes. (**C,D**) Co-expression networks derived from striosome and matrix RNA-seq data show targets of *Olig2* and *Foxf2* are co-regulated in the striosome.

*Figure 3 continued on next page*

Figure 3 continued

Genes with higher connectivity to other target genes are designated as hub genes. (E) Pathway and network enrichment (GO-BP) for the striosome (left, red) and matrix (right, blue). The resulting enrichment map was annotated using the AutoAnnotate Cytoscape App.

dramatic enrichment of consensus binding sequences recognized by bZip (Atf3, Jun, Fos) and HMG (Sox) family TFs. On the other hand, motifs recognized by Homeobox (Dlx1, Dlx2) TF family were enriched in the EGFP– OCRs (*Figure 4E–H*) The known list of TFs identified by Homer are provided as *Figure 4—source data 3*. Next, we used TOBIAS (*Bentsen et al., 2020*) to directly assess the differential occupancy and binding of TFs in striosome and matrix OCRs. The differential footprinting analysis revealed a clear enrichment for bZip family (Fos-Jun) at EGFP+ OCRs and homeobox family at EGFP– OCRs (*Figure 4I*). Annotation of 'FosJunD'-bound regions identified 365 potential genes directly regulated by this TF family. We employed ChEA software (*Lachmann et al., 2010*) to integrate our data with ChIP-chip, ChIP-seq, ChIP- PET, and DamID databases. Interestingly, potential FosJunD target genes are over-represented in Olig2-ChIP (p-value=1.482E-12), suggesting that those TFs functionally cooperate in the striosomes, in agreement with RNA-seq based network analysis. We also employed ANANSE (ANalysis Algorithm for Networks Specified by Enhancers, https://github.com/vanheeringen-lab/ANANSE; *Xu et al., 2021*) to define an atlas of key transcription factors required for cell fate determination. The top 15 TFs included *Sox10*, *Olig1*, *Olig2*, *Epas1*, *Sox8*, *Foxf2*, *Sox6*, *Nr4a1*, *Etv4*, *Ets1*, *Nkx2-2*, *Tcf7l2*, Foxc1, *and Stat3* (*Figure 4—figure supplement 1*).

## *Foxf2* deletion impairs striosome compartmentation and overexpression induces compartmentation and MSN maturation

*Foxf2*, noted above to be a hub gene (*Figure 3*) and highly enriched in the EGFP+ fraction, is a member of the *Fox* family of TFs that regulate the expression of genes involved in embryonic development, as well as in the adult. *Foxf2* is expressed in brain endothelial cells and is critical for elements of craniofacial and cochlear development (*Bademci et al., 2019*; *Hupe et al., 2017*). We confirmed that Foxf2 protein colocalizes to a large degree, but not perfectly, with DARRP-32 striosomes on PND3 (*Figure 5A*), as do the mRNAs (*Figure 5B*), and DARPP-32-independent Foxf2 vascular localization is also visualized in both. *Foxf2* mRNA and protein are undetectable in adult striatal neurons (*Figure 5—figure supplement 1A*). *Foxf2* deletion is postnatal lethal, so to determine the role of *Foxf2* in striosome compartmentation, we analyzed wild-type and *Foxf2*-null [knockout (KO)] mice at E18.5 by immunohistochemistry (IHC). Indeed, DARPP-32 and TH immunostaining revealed that striosomes were absent in the *Foxf2* KO mice (*Figure 5C*). Moreover, although the expression level of DARPP-32 was decreased (*Figure 5D*), the presence of diffuse DARPP-32-immunopositive cells in the striatum of E18.5 *Foxf2*-null mice (*Figure 5C*) suggests that the early born MSNs migrate to the striatal mantle, but other elements required for striosome organization are impacted. In fact, expression of some of the members of the Eph/Ephrin signaling system associated with control of brain cytoarchitecture (*Cooper et al., 2009*; *Dufour et al., 2003*; *Janis et al., 1999*; *Kim et al., 2016*; *Tai and Kromer, 2014*) is reduced in *Foxf2* KO mice (*Figure 5D*).

Conversely, endogenously regulated, early overexpression of *Foxf2* in a BAC transgenic (*Reyahi et al., 2015*) leads to a 40% increase in total striosomal area at PND3 (*Figure 5E,F*), and to an increase in expression in some, but not all, striosome markers (*Figure 5G*). The changes in gene expression levels are largely not maintained in the adult (*Figure 5—figure supplement 1B*), consistent with the lack of expression of *Foxf2* in the adult MSNs (*Figure 5—figure supplement 1A*). The extended overexpression of *Foxp2*, however, implies an element of a long-lasting effect on striosomes.

We also assayed the effect of Foxf2 on MSN maturation in vitro. Similar to early postnatal striatum in vivo, adenoviral-mediated overexpression of human FOXF2 promoted an increase in the percentage of DARPP-32-immunopositive cells in primary mouse neuronal cultures derived from embryonic striatum (*Figure 5—figure supplement 2A*) There was also an increase in striosome (*Oprm1*, *Foxp2*, *Rasgrp1*) and matrix (*Calb1*, *Rasgrp2*, and *EphA4*) marker mRNAs in wild-type primary striatal neurons transduced with ADV-*FOXF2*-mCherry (*Figure 5—figure supplement 2A*), again suggesting that Foxf2 likely has an effect on overall striatal development, not restricted to striosomes.

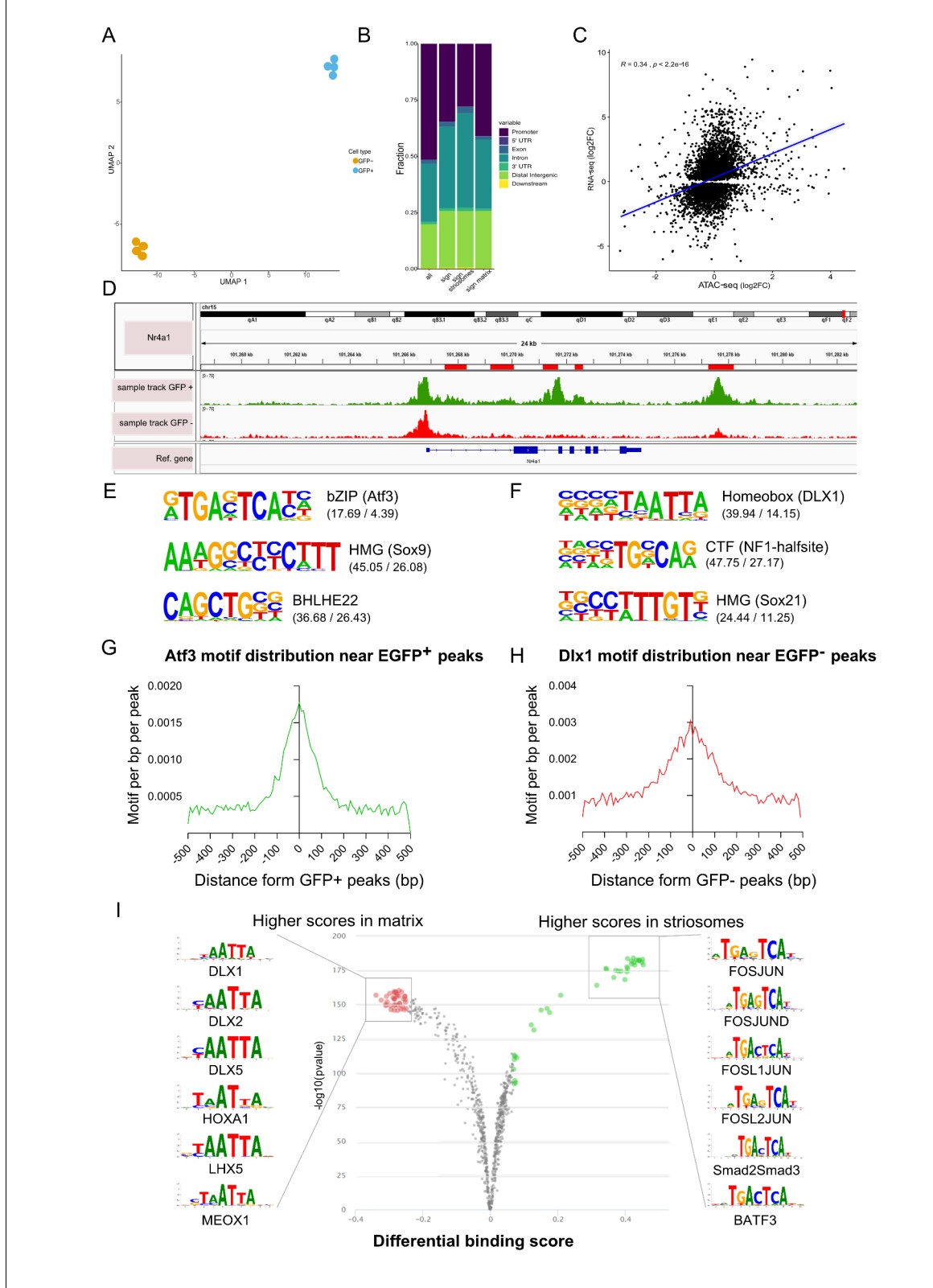

**Figure 4.** Open chromatin region (ATAC-seq) analysis identifies OCRs differentially located in striosome or matrix. (**A**) Clustering of the individual samples using UMAP. (**B**) Distribution of the location within the gene (see Materials and methods, Annotation of OCRs) of all and differentially distributed OCRs. (**C**) Correlation of log2 fold-changes in RNA-seq and ATAC-seq analyses. OCRs within TSS were considered for this analysis, and only genes with adjusted p-value <0.01 from RNA-seq analysis were included. (**D**) *Nr4a1* ATAC-seq tracks in EGFP-positive and -negative populations

*Figure 4 continued on next page*

eLife Research article

Developmental Biology | Neuroscience

Figure 4 continued

showing the presence of multiple, possible striosome 'specific' *Nr4a1* regulatory elements indicated by the red boxes in the gene annotation. (E,F) Motif analysis of OCRs in striosome (E) and matrix (F). Position weight matrix of enriched nucleotide sequence, the TF family that matches with the motif, and the predicted binding TF are shown. The values in parentheses indicate the percentage of the input sequences that contain the motif relative to the percentage of total background regions filtered by p-value of 0.01. (G, H) Frequency of predicted binding sites for Atf3 (left) and Dlx1 (right) as a function of distance from the center of striosome and matrix OCRs. (I) Pairwise comparison of TF activity between striosome and matrix. The volcano plots show the differential binding activity against the -log10(pvalue) of the investigated TF motifs; each dot represents one motif. For striosome-specific factors, the significant TFs are labeled in green, whereas matrix-specific factors are labeled in red.

The online version of this article includes the following source data and figure supplement(s) for figure 4:

**Source data 1.** Uniquely mapped reads following ATAC seq of Nr4a1-EGFP+ and Nr4a1-EGFP- cells.

**Source data 2.** Comparison of striosome (Nr4a1-EGFP+) vs matrix (Nr4a1-EGFP-) peaks representing 44% of the total number of OCRs which remained statistically significant after multiple testing corrections.

**Source data 3.** Transcription factors identified by HOMER which putatively bind to enriched binding motifs in Nr4a1-EGFP+ and Nr4a1-EGFP- cells.

**Figure supplement 1.** Analysis with ANANSE integrating RNAseq and ATACseq identifies key TFs required for striosome MSN differentiation.

## *Olig2* deletion impairs striosome compartmentation

Next, we evaluated the role of *Olig2* in the developing striosome compartment, including the validation of an associated striosome-specific OCR peak. On PND3, Olig2 protein co-localizes with DARPP-32 in striosomes, and individual Olig2-immunopositive cells devoid of DARPP-32 signal are visible throughout the striatum, likely representing oligodendroglial lineage cells (*Figure 6A*). We confirmed this distribution using two antibodies raised against Olig2, including one previously validated in *Olig2*-null mice (*Wichterle et al., 2002*; *Figure 6—figure supplement 1A*). *Olig2* mRNA also largely colocalized with *DARPP-32* mRNA in striosomes (*Figure 6B*). Note in the IHC that Olig2 protein is cytoplasmic in distribution in the striosomes, but is nuclear in the non-neuronal cells.

To determine if the ATAC-seq identified functional, compartment-specific OCRs distinct from promoters, we sought to validate a putative enhancer peak in a gene preferentially expressed in the striosomes. We compared the *Olig2* ATAC-seq tracks from EGFP+ and EGFP– cells and identified a peak 4.4 kb downstream of the *Olig2* gene (*Figure 6C*). We cloned the putative *Olig2* enhancer (chr16:91,233,041–91,234,111) into an mCherry reporter plasmid containing only the core hsp68 promoter (*Hauberg et al., 2020*; *Figure 6—figure supplement 1A*) and obtained 14 positive founders after pronuclear injection. mCherry was expressed in 9/9 founders on PND3 and co-localized with DARPP-32 neurons in the striosome (*Figure 6D*). mCherry was also expressed in individual cells in the matrix which are DARPP-32-negative (indicated by arrowheads), and probably again represent cells of the oligodendroglial lineage. The transgenic reporter was expressed in 2/5 founders assayed as adults, but its expression was low and absent in NeuN+ neurons (*Figure 6—figure supplement 1C,D*).

To determine if Olig2 is required for striosome compartmentation, we utilized a mouse in which a Cre recombinase cDNA inserted into the *Olig2* gene leads to a functional knockout (JAX 025567) (*Zawadzka et al., 2010*). Homozygote animals survived until late on the day of birth, so brains were assayed early on the day of birth. Remarkably, striosomes were absent, particularly in the anterior striatum (*Figure 6E*). DARPP-32 immunopositive neurons were essentially undetectable in the *Olig2* P0-null striatum, and although we have not yet identified the exact mechanism, the data suggest either regulation of *Ppp1r1b* gene expression by Olig2 and/or failure of striosome neurogenesis or migration. Quantitation of striosome and matrix markers in these mice revealed a marked decrease in *Ppp1r1b*, *Oprm1*, and *Foxp2* (*Figure 6F*). We also performed intracerebroventricular injection of AAV-CMV-Olig2-IRES-EGFP on P0, and analyzed the effect of Olig2 overexpression on MSN markers, and surprisingly, found a decrease in *Oprm1* and *Foxp2* mRNAs, but an increase in *Rasgrp1* (*Figure 6—figure supplement 1E,F*), indicating a mixed effect on the striosomes, which requires further exploration.

Next, we sought to determine if expression of OLIG2 in mouse primary MSNs promotes maturation toward a striosome phenotype (*Figure 6—figure supplement 2A,B*). Adenovirus-mediated expression of OLIG2 does not increase the number of DARPP-32-immunopositive cells, but importantly, it increases levels of the mRNAs for striosome markers *Oprm1*, *Foxp2* and *Rasgrp1*.

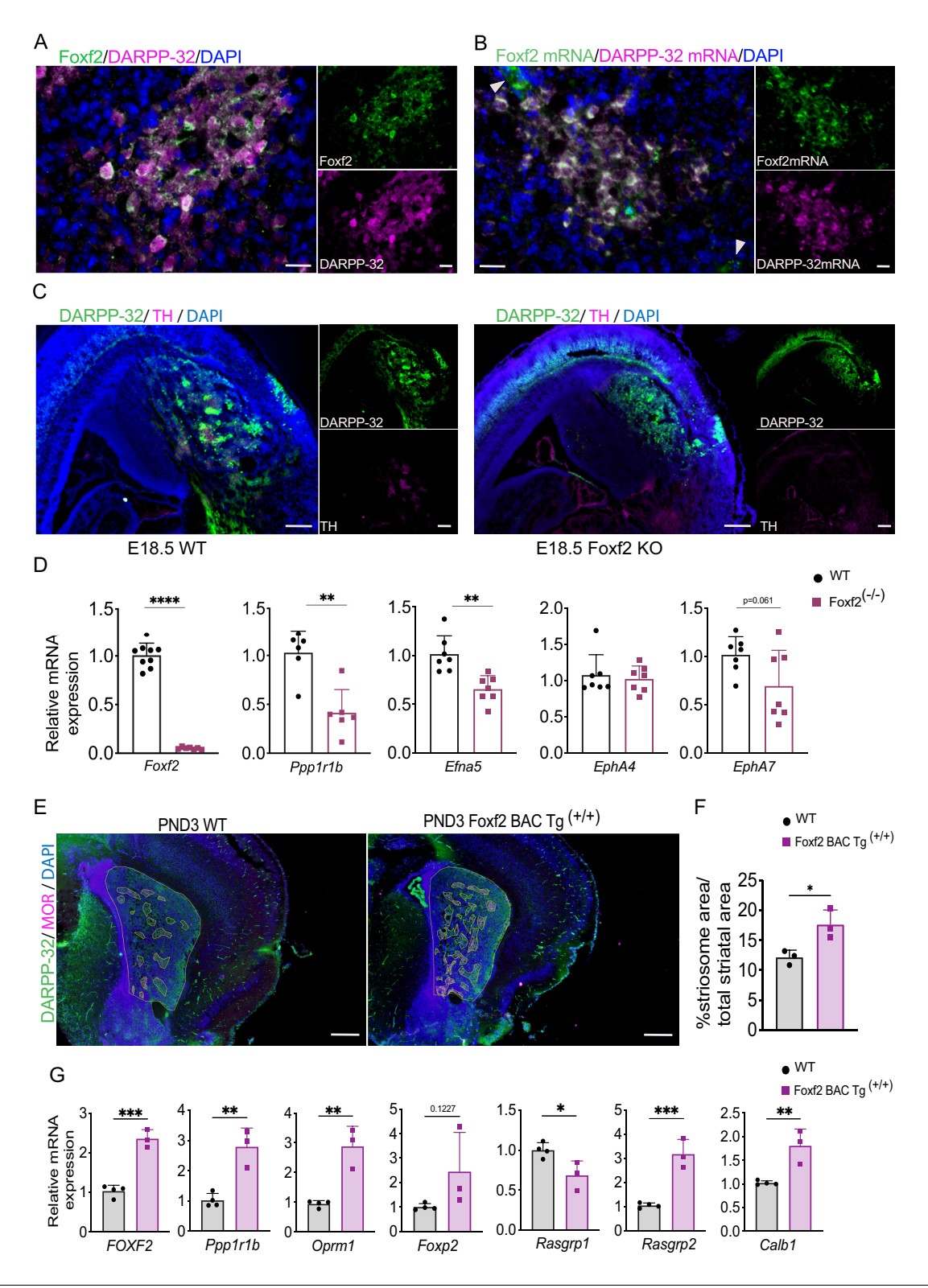

**Figure 5.** *Foxf2* is required for striosome compartmentation. (**A**) Representative confocal micrographs of Foxf2 (green), DARPP-32 (magenta) and DAPI (blue) immunolabeling on 16-μm-thick coronal sections from wild-type PND3 showing the localization of Foxf2 within the DARPP-32-immunopositive striosomes. Scale bars correspond to 20 μm. (**B**) RNAscope analysis in 16-μm-thick coronal sections from wild-type PND3 mice shows *Foxf2* mRNA (green) co-localization with *Ppp1r1b*/DARPP-32 mRNA (magenta). Arrow heads indicate vascular localization of *Foxf2* mRNA, with absence of DARPP-32

*Figure 5 continued on next page*

Figure 5 continued

mRNA. Scale bars = 20 µm. (**C**) DARPP-32 (green) and tyrosine hydroxylase (TH) (magenta) immunostaining in sagittal sections of E18.5 WT and *Foxf2* KO mice show the absence of striosome assembly in *Foxf2* KO mice. Scale bars = 100 µm (**D**) RT-qPCR from E18.5 *Foxf2* WT and null littermates striatal RNA indicating that *Foxf2* expression was abolished in the null mouse and that the levels of *Ppp1r1b and Efna5* mRNA were significantly reduced by *Foxf2* deletion. n=7 *Foxf2*$^{-/-}$ and n=9 WT. Unpaired t-test **p<0.001, ****p<0.0001. Error bars = standard deviation. (**E,F**) Comparison of area occupied by striosomes identified by merged DARPP-32 (green) and MOR (magenta) immunolabeling of 16-µm-thick coronal sections from PND3 WT and overexpressing Foxf2 (*Foxf2* BAC Tg $^{(+/+)}$) littermates showing an increase of striosome density (highlighted) mediated by Foxf2 overexpression. n=three individual mice per genotype. Unpaired t-test *p<0.05. Error bars = standard deviation. (**G**) RT- qPCR assay shows increase in the expression of both striosome (*Ppp1r1b, Oprm1, Foxp2*) and matrix (*Rasgrp2*) markers in the striatum of PND3 mice overexpressing Foxf2. n=three mice per genotype. Unpaired t-test *p<0.05, **p<0.01, ***p<0.001. Error bars represent standard deviation.

The online version of this article includes the following figure supplement(s) for figure 5:

**Figure supplement 1.** *Foxf2* mRNA is undetectable in adult striatum but Foxf2 overexpression increased the level of *FoxP2* mRNA in PND21 Foxf2 BAC Tg $^{(+/+)}$ striatum.

**Figure supplement 2.** Foxf2 overexpression in medium spiny neurons promote their maturation in vitro.

## Transcription network analysis defines the STAT pathway as a key regulator of striosome development

We used the arithmetic mean of the rank of each of the TFs at each node to quantitatively assess the importance of individual TFs in the striosome network (*Figure 3A*). The ranking was based on the number of edges expanding from each node and the number of overlapping TFs shared between nodes that interact with *Nr4a1*. Among the 58 TFs in the striosome network, the average rank of *Stat3* is 4.5, which ranks at the top. We also built the gene regulatory network, including all differentially expressed TFs, set through the regulatory data curated by ORegAnno database (*Lesurf et al., 2016*), which identified *Stat1* as the TF that regulated the greatest number of differentially expressed TFs (314 out of 811) (*Figure 7A*). Both *Stat1* and *Stat3* are enriched in the EGFP+ cells (*Figure 2—source data 1*), so together with the in-silico analyses, these data suggest that the STAT pathway is a key regulating pathway during the differentiation of striosome neurons. To test this hypothesis, we expressed human STAT1 in primary mouse MSNs. STAT1 overexpression increased the number of DARPP-32-immunopositive cells (*Figure 7B,C*) and increased the mRNA levels of striosome markers *Ppp1r1b*, *Oprm1*, *Foxp2*, *Foxf2*, and *Olig2*, as well as the matrix marker *Calb1* (*Figure 7D*). Particularly notable is the upregulation of *Foxf2* and *Olig2*, suggesting that these two TFs may be downstream of *Stat1*.

## TFs drive MSN fate in iPSCs-derived neural stem cells (NSCs)

MSN subtypes are critically needed for disease modeling, and protocols and TFs that specifically drive a striosome phenotype are just beginning to emerge (*Cirnaru et al., 2019*). To determine if the FOXF2, OLIG2, and STAT pathways are key for human development of MSNs and represent useful TFs for disease modeling of MSN subtypes, we expressed the TFs alone or in combination in NSCs derived from HD patient-induced pluripotent stem cells (*Figure 8*; *An et al., 2012*; *Naphade et al., 2017*; *Ring et al., 2015*; *Voisin et al., 2020*; *Zhang et al., 2010*). STAT1 overexpression, with or without FOXF2, increases the number of OPRM1-immunopositive cells (*Figure 8A*), with OLIG2 producing a trend in the same direction. STAT1 and OLIG2 together also increase the expression of PPP1R1B (*Figure 8A,B*). Alone, only OLIG2 increases *RASGRP1* and *EPHA4* mRNAs, but in combination, STAT1 and FOXF2 induce *RASGRP1*, although to a lesser extent. Interestingly, *RASGRP2*, a marker of the matrix, largely behaves inversely to *RASGRP1*, consistent with their expression in vivo.

OLIG2 alone also increases the matrix marker *CALB1* mRNA, but at a lower-fold than the striosome markers. Notably, in contrast to its effect in primary MSNs, FOXF2 alone increased only *RASGRP2*, and did not even increase *TUJ1*, a marker of pan-neuronal differentiation. Along with the in vivo absence of striosome compartment in the striatum despite the presence of Ppp1r1b neurons, these data may indicate an early role restricted to compartmentation and not overall phenotypic maturation as defined by expression of gene markers. In agreement with this hypothesis, FOXF2 alone failed to induce expression of *BCL11B* mRNA, a transcription factor required for striatal development (*Arlotta et al., 2008*). As expected, expression of FOXF2, OLIG2 and STAT in HD NSCs resulted in an increase expression of pan neuronal marker Nestin (*Figure 8—figure*

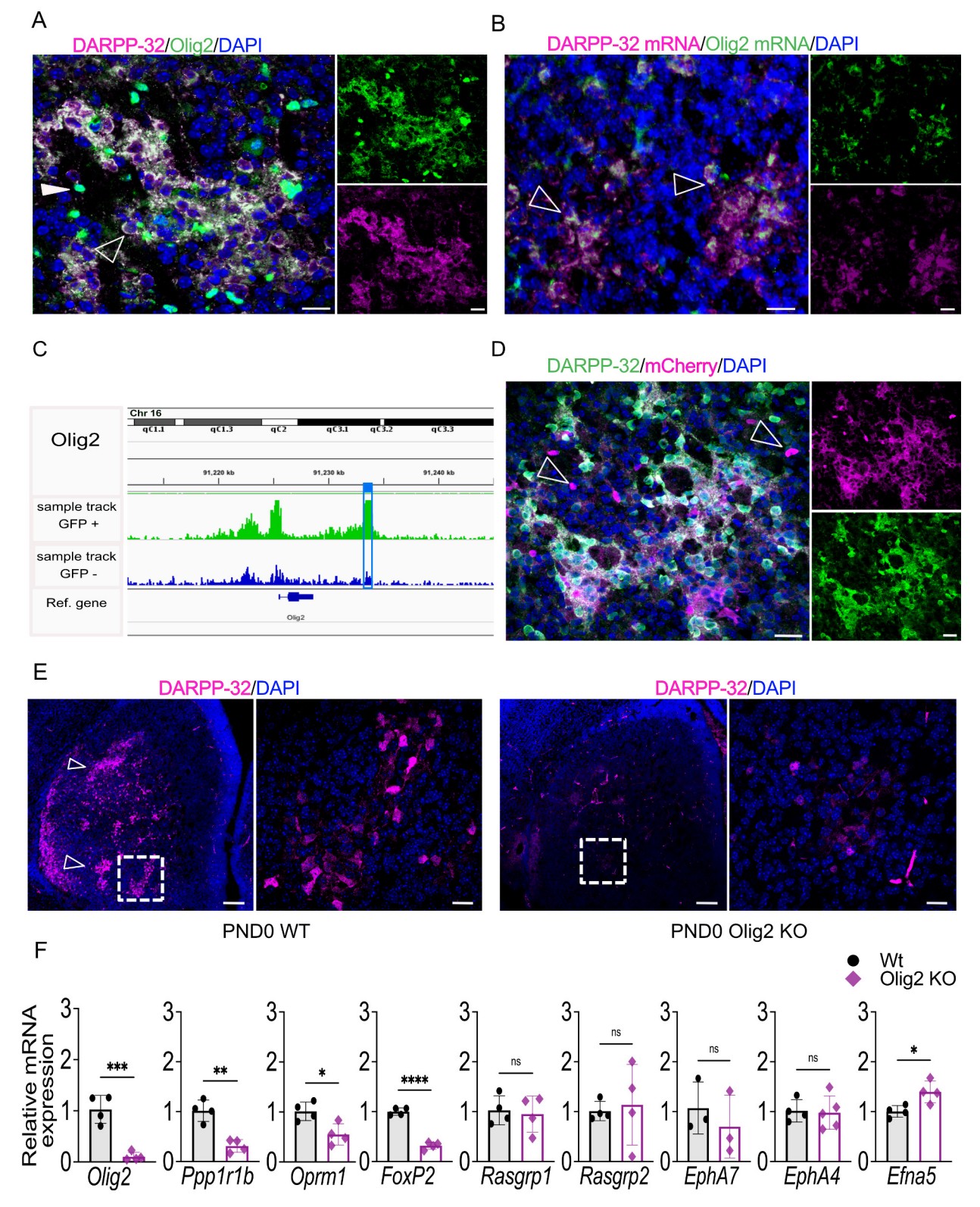

**Figure 6.** Olig2 is required for striosome compartmentation and a downstream intergenic *Olig2* OCR drives transgenic reporter expression in striosomes. (**A**) Confocal micrographs of Olig2 (green), DARPP-32 (magenta) and DAPI (blue) immunolabeling of WT PND3 16-µm-thick coronal sections showing localization of Olig2 within the DARPP-32-positive striosomes (unfilled arrowhead). An examples of a single Olig2-immunopositive and DARPP-32-negative cell is indicated by a filled overhead. Scale bars = 20 µm (**B**) RNAscope visualization of *Olig2* (green) and *Ppp1r1b/DARPP-32* (magenta)

*Figure 6 continued on next page*

*Figure 6 continued*

mRNAs showing colocalization in PND3 wild-type striosomes with individual double-labeled cells highlighted by arrowheads. (C) *Olig2* ATAC-seq tracks in EGFP$^+$ and EGFP$^-$ samples from PND3 *Nr4a1*-EGFP striatum showing the presence of an OCR 4.4 kb downstream of the *Olig2* gene, restricted to EGFP$^+$ samples (highlighted by the blue box). (D) Representative confocal micrographs of mCherry (magenta), DARPP-32 (green), and DAPI (blue) immunolabeling of 16-µm-thick coronal sections from PND3 *Olig2* -OCR transgenic mice showing that the isolated OCR sequence drives expression of the mCherry reporter restricted in large part to DARPP-32-immunopositive neurons in striosomes. Individual magenta cells, examples indicated by arrowheads, likely represent cells of the oligodendroglia lineage. Scale bars = 20 µm. (E) DARPP-32 (magenta) and DAPI immunostaining in coronal sections of PND0 WT and *Olig2* KO mice showing a dramatic reduction of DARPP-32-immunopositive neurons and striosome assembly in *Olig2* KO mice. Scale bars = 100 µm and 20 µm, respectively. (F) RT-qPCR from striatal RNA from PND0 *Olig2* KO and WT littermates indicating that striosome marker mRNAs are decreased in the KO brains. *Olig2*$^{-/-}$, n=5; WT, n=4. Unpaired t-test *p<0.05, **p<0.001, ****p<0.0001. Error bars = standard deviation.

The online version of this article includes the following figure supplement(s) for figure 6:

**Figure supplement 1.** Impact of neonatal AAV2 driven Olig2 overexpression on the striatum of PND21 mice and analysis of Olig2 OCR-driven expression in adult founders.

**Figure supplement 2.** Olig2 overexpression in medium spiny neurons promote their maturation in vitro.

supplement 1). Similar results were observed in isogenic controls C116-NSCs (*Figure 8—figure supplement 2*), in which OLIG2/STAT1 clearly induce OPRM1 expression. FOXF2, OLIG2, and STAT expression provides a rapid differentiation of DARPP-32 MSNs with high levels of MOR expression within 4 days. Current MSN protocols require 20–50 days of culture and do not generate striosomal MSN subtypes (*Arber et al., 2015*; *Golas, 2018*; *Kemp et al., 2016*; *Telezhkin et al., 2016*; *Victor et al., 2014*).

## Discussion

In the first postnatal week, striosome neurons are already compartmentalized, and their maturation is relatively advanced as compared to the matrix, based on their description as 'dopamine islands' and their expression of *Ppp1r1b*/DARPP-32 (*Fishell and van der Kooy, 1987*; *Graybiel, 1984*; *Mason et al., 2005*). We therefore hypothesized that selective sorting of *Nr4a1*-EGFP+ cells would yield young striosome neurons expressing TFs required for determination of striosome identify and low levels of markers of terminal differentiation, and the second group would be relatively immature EGFP-negative neurons committed to become matrix neurons, but not yet expressing most markers of terminal differentiation (*Figure 1*). We have previously shown a role for Nr4a1 in striosome development (*Cirnaru et al., 2019*), but based on the known expression pattern of *Nr4a1*, we were not anticipating that either set of cells would be purely neuronal. *Nr4a1* is expressed in endothelial cells (*Liu et al., 2003*), the myeloid lineage (*Wenzl et al., 2015*), OPCs (*Li et al., 2015*), and upon activation, in astrocytes (*Wu et al., 2017*).

*Foxf2* (log2FC = 7.9) and *Olig2* (log2FC = 3.05) have not previously been associated with striatal neuronal development. Foxf2 regulates the expression of Ephrin A5 and to some extent EphA7, both known to guide neuronal migration and connectivity of several brain regions, including striatum (*Cooper et al., 2009*; *Dufour et al., 2003*; *Lee et al., 2013*; *Passante et al., 2008*; *Tai and Kromer, 2014*; *Washburn et al., 2007*). In the context of gut development, components of extracellular matrix are severely reduced in the absence of *Foxf2*, leading to a lack of epithelial cell polarization and tissue assembly (*Ormestad et al., 2006*). *Foxf2* is also required to establish the cochlear cytoarchitecture (*Bademci et al., 2019*) including elements of neuronal innervation and axonal density. In vitro in primary MSNs, increased *Foxf2* levels induce genes expressed in striosomes and matrix but not in the less mature hNSCs, which together with the in vivo appearance of the *Foxf2*-null striatum, suggests that *Foxf2* alters genes associated with sorting of striosomes and matrix neurons in the embryonic period, and not necessarily with MSN-specific maturation. Notably, the vasculature may play a role in striatal compartmentation, as striosomes appear to have greater vascularization than matrix (*Breuer et al., 2005*). The importance of blood vessels in the regulation of cortical neuronal migration and maturation is well described (*Li et al., 2018*; *Paredes et al., 2016*; *Wälchli et al., 2015*), so there could be both cell-autonomous and non-cell-autonomous mechanisms via which Foxf2 regulates striatal compartmentation.

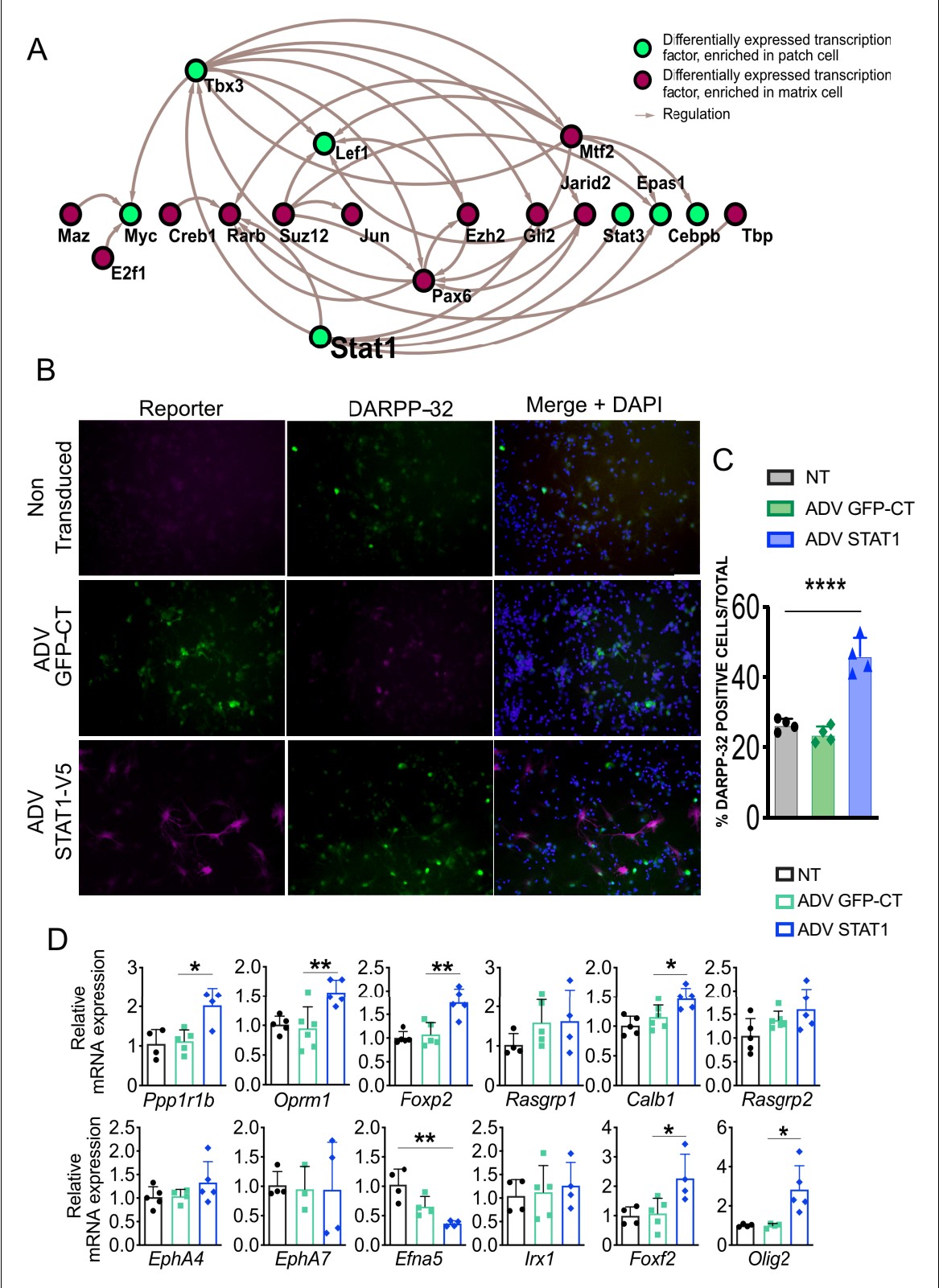

**Figure 7.** STAT1 overexpression in MSNs in vitro promotes maturation and increases levels of *Foxf2* and *Olig2* mRNAs. (A) Network analysis indicates that Stat1 may be a 'master' regulator in striosomes as it modulates the levels of the greatest number of TFs enriched in that compartment. (B,C) GFP and DARPP-32 immunolabeling in DIV9 WT primary striatal neurons either non-transduced (NT) or transduced for 96 hr with ADV-STAT1-V5 or ADV-GFP showing that STAT1 overexpression increases the number of DARPP-32-immunopositive cells. Scale bars = 50 µm. n=four images from four

*Figure 7 continued on next page*

Figure 7 continued

individual cultures, t-test ***p<0.001. (D) RT- qPCR assay shows increase of mRNA for striosome markers *Ppp1r1b*, *Oprm1*, *Foxp2*, *Foxf2*, and *Olig2*, matrix markers *Calb1* and *EphA4* and decrease in striosome enriched *EphA7* ligand and *Efna5* in DIV9 wild-type primary striatal neurons 96 hr after transduction with ADV-STAT1-V5. n=four individual cultures, one-way ANOVA corrected for multiple comparisons (Bonferroni's) *p<0.05, **p<0.01. Error bars = standard deviation.

*Olig2* (log2FC = 3.05) was not among the most enriched in its group, but both *Olig1* and *Olig2* are regulators of neuronal subtype development in the human fetal central nervous system (***Jakovcevski and Zecevic, 2005***), and in mouse, *Olig2* is regulated by *Dlx1* and *Dlx2* and appears to be involved in the neuron-glial switch (***Petryniak et al., 2007***). Specifically, *Olig2* participates in the development of GABAergic neurons in the pre-thalamus and thalamus (***Inamura et al., 2011***; ***Ono et al., 2014***; ***Wang et al., 2011a***). *Olig2* is included in the triplicated region in the Ts65Dn mouse, with marked effects on brain development in multiple regions and in iPSCs, including an increase in GABAergic interneurons (***Chakrabarti et al., 2010***; ***Xu et al., 2019***). Notably, Olig2 is highly localized to the cytoplasm in striosome neurons, similar to its localization in astrocytes where it acts to regulate differentiation (***Setoguchi and Kondo, 2004***). The validation of the predicted OCR in the *Olig2* gene as a striosome-specific enhancer in the striatum lends further support to the function of *Olig2*, while highlighting the value of the ATAC-seq database.

The use of these new data bases to construct a transcriptional regulatory network also led to the identification of *Stat1/3* as TFs able to modulate striosome and matrix phenotypes. Notably, these two members of the Stat gene family dimerize with each other (***Delgoffe and Vignali, 2013***). Neither *Stat1* nor *Stat3* has been associated with MSN development, but they are reported to promote neuronal differentiation (***Wei et al., 2014***). Further work is required to determine the role of *Stat1* in striatal development in vivo, but we did find a clear maturation effect in vitro. The effects of validated TFs on members of the ephrin family highlight the fact that, although we focused on TFs, these data sets also identify compartment-enriched mRNAs encoding proteins other than TFs that are likely critical for establishing and maintaining striatal compartmentation. For the two TFs that we validated in detail, *Foxf2* and *Olig2*, their functions do not appear to include maintenance of striosome identity, as their expression is undetectable in adult mouse MSNs, and there are no assigned striatal ATAC-seq peaks from human adult striatum (***Fullard et al., 2018***).

The transcription regulatory networks generated for each cell compartment have defined other potential TFs and pathways regulating striosome and matrix cell fate. The co-expression enrichment analysis highlighted a set of TFs that coexist with *Foxf2*, including *Sox7*, *Sox17*, *Sox18*, *PrrX1*, *Tead2*, *Bcl6*, *Klf4*, *Egr2*, and *Tbx2*. Other than *Tead2* and *Tbx2*, these TFs are enriched in the EGFP+ cells. *Sox17* belongs to the *Sox* family of TFs that regulates oligodendrocyte progenitor cell expansion and differentiation during development and repair via Notch signaling (***Chew et al., 2019***). Both *Sox17* and *Notch* signaling regulate *TCF7L2* expression (***Chew et al., 2019***), which is enriched in the same compartment. *TCF7L2* regulates calcium signaling (***Ye et al., 2020***), a function associated with both gene sets. In spinal cord patterning, *TCF7L2* in partnership with *Tcf4* represses *Olig2* expression by recruiting HDAC activity and is required for cell-fate specification (***Wang and Matise, 2016***). Histone deacetylase activity likewise plays a role in MSN maturation (***Chandwani et al., 2013***). For *Olig2*, the TF co-expression network highlights *Etv5*, *Mitf*, *Egr2*, *Sox10*, *Olig1*, *Hey1*, and *Arntl2*, which are also mostly enriched in the EGFP+ compartment. There is considerable evidence of cooperation between *Olig2* and *Olig1* in other systems, including neuronal cells (***Kim et al., 2011***). Thus, detailed analyses of these databases point to multiple TF networks likely to be involved in striosome development.

Consistent with our hypothesis, the EGFP– cells were indeed enriched in markers of immature MSNs. Most notably, these included multiple members of the *Dlx* family, required for patterning and differentiation (***Long et al., 2009***), *Isl1* which promotes commitment of striatonigral neurons (***Ehrman et al., 2013***), and markers of neuronal differentiation but not specifically related to the striatum, e.g., *Sox11* and *Pax6*, the latter of which is expressed in the ventricular zone of the lateral ganglionic eminence (***Puelles et al., 2000***). The *Arx* family is involved in neuronal progenitor proliferation and was identified by its association with lissencephaly syndromes, some of which have specific basal ganglia deficits (***Fulp et al., 2008***). *Eomes/Tbr2* is associated with cortical development, as is *Tbr1*, raising the possibility of contamination by cortical elements during dissection and

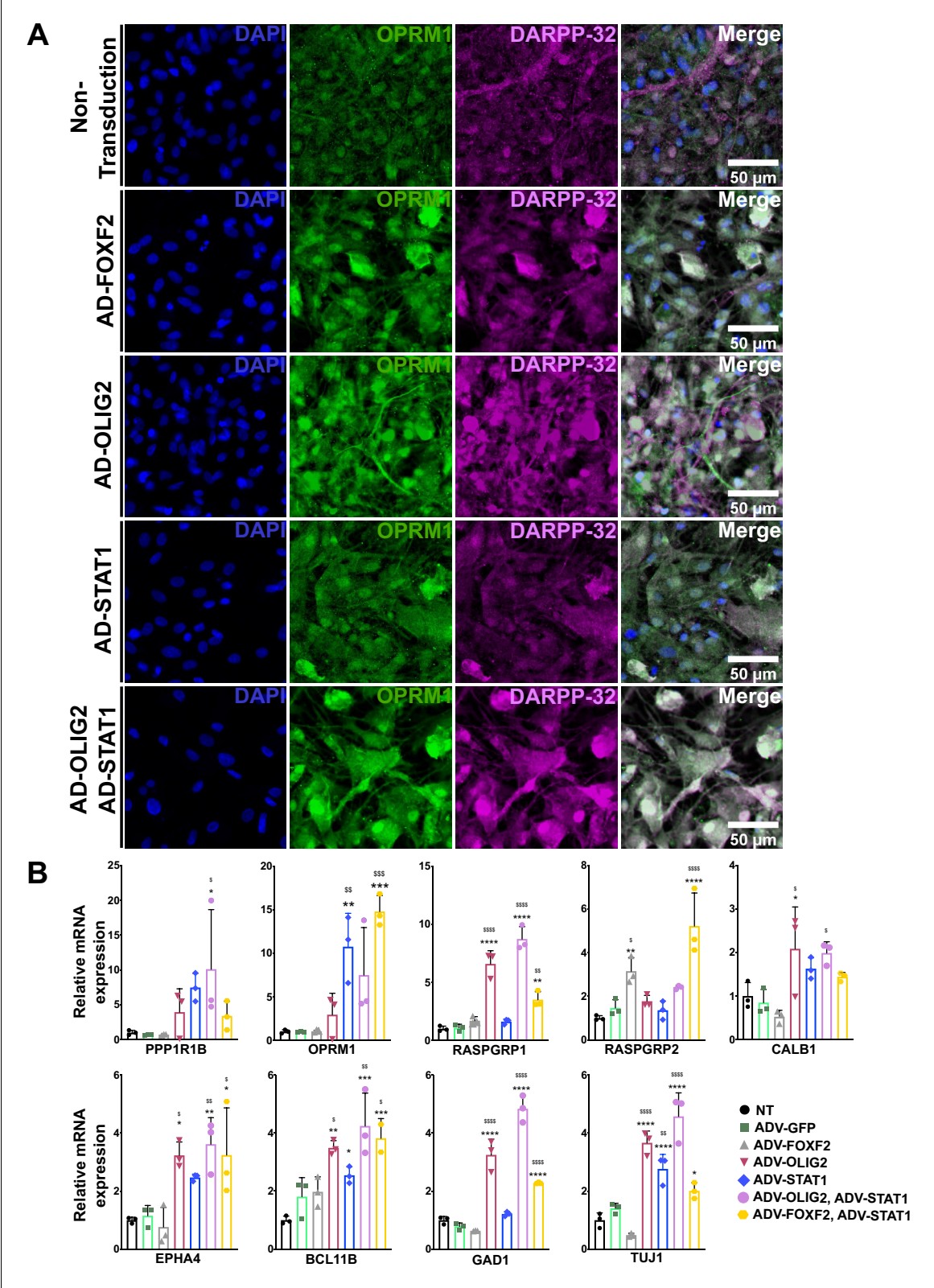

**Figure 8.** FOXF2, OLIG2, and STAT1, alone and in combination, promote MSN differentiation in NSCs from human HD induced pluripotent stem cells. (**A**) HD72-NSCs transduced for 4 days with ADV-*FOXF2*, ADV-*OLIG2* and ADV-*STAT1* were immunostained with Oprm1 (green) and DARPP-32 (magenta). Non-transduced cells and AD-GFP were used as control. Scale bars = 100 µm. (**B**) RT-qPCR assay on HD72-NSCs transduced for 4 days with ADV-GFP, ADV-*FOXF2*, ADV-*OLIG2* and ADV-*STAT1*. n=three individual cultures. One outlier (one of the three technical data points) was excluded in

*Figure 8 continued on next page*

*Figure 8 continued*

*Bcl11b* RT-qPCR when transduced with ADV- *FOXF2* and ADV-*STAT1*. One-way ANOVA for multiple comparisons (Dunnett's) \*p<0.05, \*\*p<0.01, \*\*\*p<0.001, \*\*\*\*p<0.0001. \* for comparison to non-transduced and for comparison to ADV-GFP. Error bars are standard deviation.

The online version of this article includes the following figure supplement(s) for figure 8:

**Figure supplement 1.** Expression of FOXF2, OLIG2 and STAT in HD NSCs resulted in an increase expression of pan neuronal marker Nestin in HD NSCs.

**Figure supplement 2.** FOXF2, OLIG2, and STAT1, alone and in combination, promote MSN differentiation in NSCs from human isogenic control C116-induced pluripotent stem cells.

cell selection (*Puelles et al., 2000*). Notably, however, both *Tbr1* and *Satb2* mRNAs are detected in the striatum [Allen Mouse Brain Atlas (mouse.brain-map.org)]. Interestingly, *Foxp2* and *Oprm1* are associated with striosomes (*Campbell et al., 2009*) but are enriched in the EGFP– cells. These data imply that, in some cases, transcription and translation may be dissociated at this stage. Indeed, *Oprm1* mRNA distribution is initially diffuse during striatal development (*Tong et al., 2000*) and the Allen Brain Atlas fails to confirm a striosome distribution for *Foxp2* at this age.

In summary, we present two data bases derived from PND3 striatum that compare gene expression and OCRs in developing striosome and matrix, demonstrating clear gene expression and epigenetic distinctions. Using these data sets, we describe a novel pathway via which Stat1 appears to regulate a transcriptional hierarchy that includes Foxf2 and Olig2, critical for striosome compartmentation and phenotypic maturation. Further, we demonstrate that these TFs can be utilized to better model striatal striosome MSNs in hiPSC systems and to develop genetic tools to direct expression to neuronal subsets for their eventual in vivo manipulation and study.

# Materials and methods

## Animals

Animal procedures were conducted in accordance with the NIH Guidelines for the Care and Use of Experimental Animals and were approved by the Institutional Animal Care and Use Committee of our institutions (LA09-00272, 16–0847 PRYR1). The *Nr4a1*-EGFP mice used for this study were obtained from GENSAT and the Olig2-Cre mice were obtained from Jackson Laboratories. All recombinant *Foxf2* mice were kindly provided by Dr. Peter Carlsson. Mice were given ad libitum access to food and water and housed under a 12 hr light/dark cycle. Both male and female mice were used in these studies.

## Enzymatic dissociation and FACS purification of striatal neurons

PND3 *Nr4a1*-EGFP heterozygous mice were obtained from a time set homozygote *Nr4a1*-EGFP bred to WT pairs. The pups were separated by sex and rapidly euthanized by decapitation, and the striata dissected under the microscope in ice-cold Hibernate-A medium (A1247501, Gibco, Thermo Fisher). The striata were collected from three PND3 mice per sample and exposed to enzymatic dissociation with papain (*Lobo et al., 2006*) with minor modification. Briefly, the tissue was incubated with frequent agitation at 37°C for 20 min in 2 ml of 2 mg/ml papain (Sigma, P4762) solution in Hibernate-A. The tissue was washed twice with 5 ml of Hibernate-A, including 1 mg/ml protease inhibitor (Thermo Fisher, 78442), and briefly triturated in 2 ml of Hibernate-A medium using fire-polished glass Pasteur pipettes until a single-cell suspension was obtained. The cell suspension was filtered through 70 µm mesh previously equilibrated with 2 ml of Hibernate-A. The cells left on the filter were collected by washing the filter with 1 ml of Hibernate-A. The cells were centrifuged for 5 min at 700 g and treated for 10 min with DAPI (4′,6-diamidino-2-phenylindole) (1:4000, Sigma, 62248)to label dead cells. The cells were washed with 10 ml of Hibernate-A, centrifuged for 5 min at 700 g and resuspended in 0.35 ml of Hibernate-A. Viable cells (DAPI negative) were sorted and collected into two populations: EGFP+ and EGFP– using a BD Influx FACS sorter based on fluorescein-5-isothiocyanate (FITC) channel signal for EGFP. WT striatal cells were used to calibrate the FITC and DAPI signals. The cells for RNA-seq (1000 cells/sample) were collected in 150 µl of Arcturus PicoPure extraction buffer. After the sorting, the samples were incubated at 42°C for 30 min and stored at −80°C until RNA extraction was performed using Arturus PicoPure RNA Isolation kit

(Applied Biosystems, 12204–01). The cells for ATAC-seq (25,000 cells/sample) were collected in 1.5 ml Eppendorf tubes coated with 5% BSA, centrifuged at 300 g for 10 min at 4℃, and the pellet was stored.

## Tissue preparation and immunofluorescence

PND3 mice were rapidly euthanized by decapitation, and brains were removed, washed in ice-cold PBS, and post-fixed for 24 hr at 4℃ in 4% PFA. The brains were then incubated in 30% sucrose/1X PBS for 24 hr at 4℃ and cryopreserved in OCT embedding medium (4583, Tissue-Tek Sakura). Serial coronal sections (16 µm) were cut on a Leica cryostat, collected on Superfrost Plus microscope slides (Fisher Scientific) and frozen at −20℃. Immunofluorescence was performed as described (*Cirnaru et al., 2019*). Sections were incubated with mouse anti-DARPP-32 (1:250, sc-271111, Santa Cruz Biotechnology), sheep anti-Foxf2 (1:2000, AF6988, R and D), rabbit anti-Olig2 (1:500, ab136253, Abcam), rabbit anti-Olig2 (1:8000, gift from Dr. Wichterle) (*Wichterle et al., 2002*), mouse anti-Olig2-Alexa488 (1:1000, MABN50A4, Millipore) rabbit anti-Irx1 (1:2000, PA5-36256, Thermo Fisher Scientific) or rabbit anti-tyrosine hydroxylase (1:1000, OPA1-04050, Thermo Fisher Scientific) antibodies. The respective secondary antibodies included: anti-mouse Alexa 488 (1:400, A-11008, Thermo Fisher Scientific), anti-mouse Alexa 594 (1:400, A-11005, Thermo Fisher Scientific), anti-rabbit Alexa 488 (1:400, A-11034, Thermo Fisher Scientific), anti-rabbit Alexa 594 (1:400, A-11012, Thermo Fisher Scientific), or anti-sheep Alexa 594 (1:400, A-11016, Thermo Fisher Scientific). Sections were sealed with Vectashield hard-set mounting medium (H-1400, Vector Laboratories). Images were acquired using an Olympus BX61 microscope or a Confocal Zeiss LSM 510.

## Fluorescent in situ hybridization (FISH) using RNAscope technology

Postnatal day 3 WT brains were fixed in freshly prepared, ice-cold 4% PFA for 24 hr at 4℃, followed by equilibration in 10% sucrose gradient, then 20% and finally 30%, each time allowing the tissue to sink to the bottom of the container. The tissue was embedded in OCT compound (4583, Tissue-Tek Sakura) and stored at −80℃ until sectioned. 16 µm-thick sections were cut using a Leica cryostat, collected onto Superfrost Plus slides maintained at −20℃ during the sectioning. The slides were then stored at −80℃. RNAscopeProbe murine Mm-DARPP-32-C1 Ppp1r1b (NM_144828.1, bp590-1674, 405901), Mm-Olig2-C3 (NM_016967.2, bp865-2384, 447091-C3)and Mm-Foxf2-C2 (NM_010225.2, bp846-2316, 473391-C2) were purchased from Advanced Cell Diagnostics probe catalog (ACD). For signal detection, we used Opal 520 and Opal 690 TSA plus fluorophores (Akoya Biosciences). The RNAscope Multiplex Fluorescent Reagent Kit v2 (ACD, 323100) used here provides the target retrieval solution, hydrogen peroxide, protease III, amplification reagents (Amp1-3), HRP reagents, DAPI, TSA buffer and wash buffer. We used a modified version of the manufacturer's protocol for sample preparation, probe hybridization, and signal detection. Briefly, the fresh frozen sections on slides were retrieved from −80℃ and briefly immersed in 1X PBS to wash off the O.C.T and then baked at 60℃ for 30 min. Slides were then post-fixed in fresh 4% PFA for 1 hr at room temperature (RT). After fixation, the sections were dehydrated in a series of ethanol solutions (5 min each in 50%, 70%, and two changes of 100% ethanol) at RT and left to dry for 5 min at RT. Sections were treated with hydrogen peroxide for 10 min at RT and washed twice with distilled water. Subsequently, target retrieval was performed by boiling the slides for 5 min in 1X Target Retrieval Reagent (ACD), washed in distilled water, immersed in 100% ethanol, and air-dried for 5 min at RT. A hydrophobic barrier was created around the section using an ImmEdge Pen (ACD, 310018) and completely dried at RT before proceeding to the next step. Sections were then treated with protease III for 5 min at 40℃ in the pre-warmed ACD HybEZ II Hybridization System (ACD, 321721) inside the HybEZ Humidity Control Tray (ACD, 310012) and washed twice with distilled water. The *Foxf2-C2* or *Olig2*-C3 probes were diluted at 1:50 in *DARPP32*-C1 probe. The sections were then hybridized with the probes, *DARPP-32* and *Olig2* or *DARPP-32* and *Foxf2*, at 40℃ for 2 hr in the HybEZ Oven (ACD), washed twice with 1X wash buffer, and stored overnight at RT in 5x SSC buffer (Thermo Fisher Scientific). The next day, the slides were rinsed twice with wash buffer for 2 min each, followed by the three amplification steps (AMP 1, AMP 2, and AMP 3 at 40℃ for 30, 30, and 15 min respectively, with two washes with wash buffer after each amplification step). The signal was developed by treating the sections in sequence with the horseradish peroxidase (HRP) reagent corresponding to each channel (e.g., HRP-C1) at 40℃ for 15 min, followed by the TSA Plus fluorophore

assigned to the probe channel (Opal 690 for DARP32-C1 probe at 1:2000 dilution and Opal 520 for *Foxf2*-C or *Olig2*-C3 probes, prepared at a dilution of 1:1500) at 40℃ for 30 min, and HRP blocker at 40℃ for 15 min, again with two wash steps after each of the incubation steps. Finally, the slides were counterstained with DAPI for 30 s, mounted using ProLong Gold mounting medium (Thermo Fisher Scientific) and stored at 4℃ until ready for imaging. The sections were imaged using a 10x/ 0.3 N.A., 20x/0.8 N.A. or 40x/0.75 N.A. objectives on an AxioImager Z2 microscope (Carl Zeiss), equipped with a Zeiss Axiocam 503, and operated with Zeiss Zen Blue software (Carl Zeiss). Camera exposure times were set for all three channels (red for Opal 690, blue for DAPI and green for Opal 520) and were identical for all slides within each experiment. ImageJ was used for adjusting the brightness and contrast of the images.

### Primary neuronal cultures

E16.5 embryos were obtained from wild-type (WT) Swiss Webster timed bred females purchased from Charles River. E16.5 striatum was removed from by microdissection in cold Leibovitz's medium (L-15) (Gibco-Invitrogen,11415064) and primary medium spiny neuronal cultures were prepared as described (*Cirnaru et al., 2019*). Briefly, the tissue was incubated in $Ca^{2+}/Mg^{2+}$-free Hanks' balanced salt solution (Sigma, 55021C) for 10 min at 37℃. The incubation mixture was replaced with 0.1 mg/ml papain in Hibernate E/$Ca^{2+}$ (BrainBits), incubated for 8 min, and rinsed in Dulbecco's minimum essential medium (Gibco-Invitrogen, 21013024) with 20% fetal bovine serum (Gibco-Invitrogen, 10438026) and twice in Leibovitz's medium (L-15). The tissue was then suspended in Dulbecco's minimum essential medium with 10% fetal bovine serum, glucose (6 mg/mL) (Sigma, G7021), glutamine (1.4 mM) (Gibco- Invitrogen, 25030081) and penicillin/streptomycin (100 U/mL) (Gibco-Invitrogen, 15140122). Cells were triturated through a glass bore pipette and plated onto either Lab Tek eight-well slides (1.25 x105 cells/well) for immunocytochemistry or 24-well plates (1x106 cells/well) for RT-PCR analysis, each coated with polymerized polyornithine (0.1 mg/mL in 15 mM borate buffer, pH 8.4) and air-dried. One hr later, the medium was replaced with Neurobasal (Gibco-Invitrogen, 21103049), supplemented with B27, (Gibco-Invitrogen, 17504044), GLUTAMAX (Gibco- Invitrogen, 35050061) and penicillin/streptomycin. The medium was changed every 2 days, and the cells were assayed on day in vitro (DIV) 9.

### Neuronal adenovirus (ADV) transduction

ADV-CMV-*OLIGO2*-mCherry, ADV-CMV-*FOXF2*- mCherry, ADV-CMV-*STAT1*-V5, and ADV-CMV-EGFP were produced by SignaGen Laboratories using the human cDNA sequence. Viral transduction was performed at DIV5 with a multiplicity of infection (MOI) of 20. The virus was added in fresh medium, and the medium was changed 18 hr later. Cells were harvested or fixed 96 hr after addition of virus.

### Mouse neuronal immunocytochemistry

Cells were fixed in 4% paraformaldehyde in 0.1 M phosphate buffer, pH 7.4, and immunolabeled with mouse anti-DARPP-32 (1:250, sc-271111, Santa Cruz Biotechnology), rabbit anti-Olig2 (1:500, ab136253, Abcam) or rabbit anti-STAT1 (1:400, 14994S, Cell Signaling Technology) followed by anti-mouse Alexa 488 (1:400, A-11008, Thermo Fisher Scientific) and anti-rabbit Alexa 594 (1:400, A-11012, Thermo Fisher Scientific). To identify the total number of cells the nuclei were stained with DAPI (4', 6-diamino-2-phenylindole dihydrochloride) (1:10,000, Millipore-Sigma). Images were acquired using Olympus BX61 microscope and analyzed using Fiji software (ImageJ).

### Real-time qPCR of cultured mouse neurons

RNA from DIV9 primary MSNs was extracted with the miRNeasy micro kit (Qiagen), according to the manufacturer's instructions. RNAs (500 ng) were reversed-transcribed using the High Capacity RNA-to-cDNA Kit (Applied Biosystems). Real-time qPCR was performed in a Step-One Plus system (Applied Biosystems) using All-in-One qPCR Mix (GeneCopoeia). Quantitative PCR consisted of 40 cycles, 15 s at 95℃ and 30 s at 60℃ each, followed by dissociation curve analysis. The ΔCt was calculated by subtracting the Ct for the endogenous control gene GAPDH from the Ct of the gene of interest. Mouse primer sequences are listed in *Table 2*. Relative quantification was performed using the ΔΔCt method and expressed as a fold-change relative to control by calculating 2-ΔΔCt.

**Table 2.** RT-qPCR murine primer sequences.

| Gene | Primer forward 5'–3' | Primer reverse 5'–3' |
| --- | --- | --- |
| GAPDH | AACGACCCCTTCATTGACCT | TGGAAGATGGTGATGGGCTT |
| Ppp1r1b | GAAGAAGAAGACAGCCAGGC | TAGTGTTGCTCTGCCTTCCA |
| Oprm1 | CCCTCTATTCTATCGTGTGTGT | AGAAGAGAGGATCCAGTTGCA |
| Foxp2 | AAGCAGCTTGCCTTTGCTAAG | GGATTGAATGTATGTGTGGCTGA |
| Rasgrp1 | GGACCTACCAAGAACTGGAAC | GATCCCAGTAAACCCGTCTG |
| Calb 1 | ACTCTCAAACTAGCCGCTGCA | TCAGCGTCGAAATGAAGCC |
| Rasgrp2 | CTTGGACCAGAACCAGGATG | GTGGCAGTTCACACCACAAG |
| Epha7 | TGCTCCGCTTTGCACACACAGG | TAAGTTCTCAATAATGGACCAGCAC |
| Epha4 | TCGTGGTCATTCTCATTG | TCTCTTCATCTGCTTCTTG |
| Ephna5 | CGATAGAACCAAGATAATACT | TAGAATCAGAGGACTCAG |

## Human-induced pluripotent stem-cell-derived NSC culture

All work on human iPSCs was approved by the Buck Institute institutional stem cell and the human ethics committee (Approval S1002). The isogenic C116 and HD iPSCs were previously published (*An et al., 2012*; *Naphade et al., 2017*; *Ring et al., 2015*; *Voisin et al., 2020*; *Zhang et al., 2010*), were sequenced, karyotyped and were negative for mycoplasma contamination. Human-induced pluripotent stem cells (iPSCs) were differentiated into prepatterned activin A-treated neural stem cells (NSCs) by the following protocol. Briefly, iPSC colonies were detached using 1 mg/ml collagenase (Type IV, Thermo Fisher Scientific, 17104019) in Gibco KnockOut DMEM/F-12 medium (Thermo Fisher Scientific, 12660012), and the resulting cell clumps were transferred to a 0.1% agarose (Sigma- Aldrich, A9414) coated low-attachment petri dish in embryonic stem (ES) culture medium Gibco KnockOut DMEM/F12 supplemented with 20% Gibco KnockOut Serum Replacement (Thermo Fisher Scientific, 10828028), 2.5 mM L-glutamine (Thermo Fisher Scientific, 25030081), 1 X Non-Essential Amino Acids (NEAAs) (Thermo Fisher Scientific, 11140050), 15 mM HEPES (Thermo Fisher Scientific, 15630106), 0.1 mM β-mercaptoethanol (Thermo Fisher Scientific, 31350010), 100 U/ml penicillin-streptomycin (Thermo Fisher Scientific, 15140122). Every 2 days, 25% of ES medium was replaced by embryoid body (EB) differentiation medium [DMEM (Corning, 10–013- CV) supplemented with 20% FBS (Thermo Fisher Scientific, 16000036), 1 X NEAA, 2 mM L- glutamine, 100 U/ml penicillin-streptomycin]. At day 8, 100% of the culture medium was EB medium. At day 10, the embryoid bodies were attached to dishes coated with poly-L-ornithine (1:1000 in PBS; Sigma-Aldrich, P3655) and laminin (1:100 in KnockOut DMEM/F-12; Sigma- Aldrich, L2020), and cultured in neural induction medium [DMEM/F12 supplemented with 1 X N2 (Thermo Fisher Scientific, 17502001), 100 U/ml penicillin-streptomycin] and 25 ng/ml βFGF (Peprotech, 100-18B) and 25 ng/ml Activin A (Peprotech, 120–14P). Media change was performed every 2 days. Rosettes were harvested after 7–10 days, plated on poly-L-ornithine- and laminin-coated dishes, and cultured in Neural Proliferation Medium [NPM; Neurobasal medium (Thermo Fisher Scientific, 21103049), B27-supplement 1 X (Thermo Fisher Scientific, 17504001), GlutaMAX 1 X (Thermo Fisher Scientific, 35050061), 10 ng/ml leukemia inhibitory factor (Peprotech, 300–05), 100 U/ml penicillin-streptomycin] supplemented with 25 ng/ml β-FGF and 25 ng/ml activin A. The resulting NSCs were passaged and maintained in this same medium. These prepatterned activin A-treated NSCs were validated by immunofluorescence analysis and labeled positively for putative NSC markers, namely Nestin, SOX1, SOX2, and PAX6.

## Adenovirus transduction of human NSCs

For the ADV transduction experiments human iPSC- derived NSCs were plated at 700,000 cells per well of a six-well plate in 2 ml of NPM supplemented with 25 ng/ml βFGF (Peprotech, 100-18B) and 25 ng/ml activin A (Peprotech, 120–14P). Two day after plating, cells were transduced at MOI of 10 with ADV-CMV-*FOXF2*-mCherry (SignaGen Laboratories, SL100701), ADV-CMV-OLIG2-mCherry (SignaGen Laboratories, SL100756) and ADV-CMV-STAT1-6xHN (SignaGen Laboratories, SL110858) and MOI of 0.75 with ADV-CMV-EGFP (SignaGen Laboratories, SL100708) in 1 ml of NPM without

penicillin/streptomycin. Then, the cells were placed on an orbital shaker in 37°C for 1 hr. A complete media change was performed 24 hr post-transduction. Non-transduced NSCs and NSCs transduced with ADV-EGFP were used as controls. Cells were harvested 4 days after transduction for gene expression and immunolabeling assays. ADV transduction in a human iPSC-derived NSC culture was performed twice, with three replicates each.

## Cell immunofluorescence of human NSCs

Cells were fixed using 4% paraformaldehyde in 0.1 M phosphate-buffered saline (PBS), pH 7.4 (Corning, 21–040-CV) for 30 min. After three washes in PBS, cells were permeabilized and blocked for 1 hr at RT using 0.1% Triton X-100 (Thermo Fisher Scientific, 28313) and 4% donkey serum in PBS. Primary antibodies were added in the presence of blocking buffer overnight at 4°C. Secondary antibodies (1:500) were added after three PBS washes in blocking buffer at RT for 1 hr. The following primary antibodies were used for the immunofluorescence studies: rabbit anti-DARPP-32 (Santa Cruz, sc-11365, 1: 100) and rabbit anti-Opioid Receptor-Mu (Millipore, AB5511, 1:500). The secondary antibodies were donkey anti- rabbit IgG conjugated with Alexa-488 (Invitrogen, A12379) or Alexa-647 (Invitrogen, A22287). Images were acquired using a Biotek Cytation five microscope and were prepared using Fiji software (ImageJ).

## Quantitative real-time PCR of human NSCs

For qRT-PCR analysis of prepatterned activin A- treated human NSCs, total RNA was isolated using the ISOLATE II RNA Mini Kit (Bioline, BIO- 52072). cDNA was prepared from 300 ng of RNA in a total reaction volume of 20 μl using the Sensi-FAST cDNA synthesis kit (Bioline, BIO-65053). RT-PCR reactions were set up in a 384-well format using 2X SensiFAST Probe No-ROX Kit (Bioline, BIO-86005) and 1 μl of cDNA per reaction in a total volume of 10 μl. RT-PCR was performed on the Roche LightCycler 480 instrument. Quantitative PCR consisted of 40 cycles, 15 s at 95°C and 30 s at 60°C each, followed by dissociation curve analysis. The ΔCt was calculated by subtracting the Ct for the endogenous control gene β-actin from the Ct of the gene of interest. Human primer sequences are listed in *Table 3*. Relative quantification was performed using the ΔCt method and is expressed as a fold- change relative to control by calculating 2- ΔCt.

## Generation of RNA-seq libraries

For RNA-seq, EGFP-positive and -negative cells were sorted into low-binding tubes containing Arcturus PicoPure Extraction buffer. RNA was isolated in accordance with the PicoPure RNA Isolation kit manufacturer's instructions, which included a DNase treatment step. Samples were eluted in RNase-free water and stored at −80°C until preparation of RNA-sequencing libraries using the Takara Clontech Laboratories SMARTer Stranded Total RNA-Seq Pico Kit, according to the manufacturer's instructions. After construction of the RNA-seq libraries, libraries were analyzed on an Agilent High Sensitivity D1000 TapeStation, and quantification of the libraries was performed using the KAPA Library Quantification Kit.

**Table 3.** qRT-PCR human primer sequences.

| Gene | Primer forward 5′−3′ | Primer reverse 5′−3′ | Universal probe |
|---|---|---|---|
| *PPP1R1b* | CACACCACCTTCGCTGAAA | GAAGCTCCCCCAGCTCAT | 82 |
| *OPRM1* | AGAAACAGCAGGAGCTGTGG | ACCGAGACTTTTCGGGTTC | 30 |
| *BCL11b* | CCCAGAGGGAGCTCATCAC | TTTGACACTGGCCACAGGT | 45 |
| *CALB1* | CACAGCCTCACAGTTTTTCG | CCTTTCCTTCCAGGTAACCA | 36 |
| *GAD1* | ATGGTGATGGGATATTTTCTCC | GCCATGCCCTTTGTCTTAAC | 46 |
| *TUJ1* | GCAACTACGTGGGCGACT | CGAGGCACGTACTTGTGAGA | 78 |
| *RASPGRP1* | GAGCCAAAGATCTGCTCCAT | GGTCCGATCCTTACTCTCCTC | 71 |
| *RASPGR2* | TGAGCCACAGCTCCATCTC | CCGTCACTAGTTCCGTGAGAC | 75 |
| *EPHA4* | AGCAGCCACTCAGGCAAC | ACGAAAATAGGGCGAAATAGAA | 51 |
| *ACTB* | CCAACCGCGAGAAGATGA | CCAGAGGCGTACAGGGATAG | 64 |

## RNA-seq

RNA-seq was carried out at New York University using Illumina HiSeq 4000 Paired-End 150 Cycle Lane from purified RNA PicoPure RNA Isolation Kit KIT0204 Arcturus (ThermoScientific) and SMARTer Stranded Total RNA-Seq Kit - Pico Input Mammalian (250 pg–10 ng RNA) (635005, Clontech). The low-quality base (quality score lower than 20), as well as the adapters of the raw reads from the sequencing experiments, was removed using Trim Galore! (https://www.bioinformatics.babraham.ac.uk/projects/trim_galore/). External and internal rRNA contamination was filtered through SortMeRNA 2.1b (*Kopylova et al., 2012*). Then the filtered raw reads were then mapped to the Genome Reference Consortium Mouse Build 38 striosome release 6 (GRCm38.p6) assembly by GENCODE using STAR 2.7.2b (*Dobin et al., 2013*). The counts of reads mapped to known genes were summarized by featureCounts, using GENECODE release M22 annotation (GSE143276). The data discussed in this publication have been deposited in NCBI's Gene Expression Omnibus GEO Series accession number, GSE143276. Next, R Bioconductor package DESeq2 (*Love et al., 2014*) was used to normalize raw read counts logarithmically and perform differential expression analysis. Differentially expressed genes were based on an arbitrary cutoff of adjusted p-value less than 0.01 (*Figure 2—source data 1*). We found the regulation pattern of a gene with the same EGFP status is more likely to be the same across replicates, whereas it is more likely to be different when the EGFP status is different between replicates with the exception of *Xist*. *Xist* is a gene highly expressed in females only. The regulation pattern of *Xist* is consistent with gender differences of the samples confirming the identity of each sample (*Figure 2e*).

## Terminology enrichment analysis and pathway enrichment analysis

Enrichment analysis was performed on gene clusters in specific databases to determine if a specific biological annotation could be considered as significantly represented under the experiment result. Both terminology enrichment analysis and pathway enrichment analysis were conducted by clusterProfiler (*Yu et al., 2012*), a Bioconductor package. In our analysis, biological process (BP), molecular function (MF), and cellular component (CC) terms in gene ontology (GO) (*The Gene Ontology Consortium, 2017*), as well as pathway annotations derived from Kyoto Encyclopedia of Genes and Genomes (KEGG), were chosen to identify predominant biological processes of the differentially expressed gene clusters and differentially expressed transcription factor clusters involved in the development of the striosome neurons. We conducted both analyses on the differentially expressed gene clusters with the arbitrary cutoff of adjusted p-value less than 0.01 and the absolute Log2 foldchange greater than 0, 1, and 2, respectively, and we conducted both analyses on the differentially expressed transcription factor clusters with the arbitrary cutoff of adjusted p-value less than 0.01 and the absolute Log2 fold- change greater than 0 and 1, respectively (*Figure 2—source data 1*).

## TF enrichment analysis and co-expressor enrichment analysis

Both TF enrichment analysis and co-expressor enrichment analysis were conducted by Enrichr (*Kuleshov et al., 2016*), a comprehensive online tool for doing enrichment analysis with a variety of biologically meaningful gene set libraries (*Figure 2—source data 1*). In our analysis, ChEA (*Lachmann et al., 2010*) and ENCODE (*Frankish et al., 2019*) databases were chosen to identify the significant upstream TFs regulating genes and other TFs differentially expressed in striosome cells and matrix cells, respectively, and the ARCHS4 database was chosen to identify the significant co-expressors of those differentially expressed genes and transcription factors. An arbitrary cutoff of adjusted p- value less than 0.01 and the absolute log2 fold-changes greater than 0 and 1 were chosen (*Figure 2—source data 1*).

## GeneMANIA gene regulatory network analysis

GeneMANIA (*Warde-Farley et al., 2010*) is an online tool using published and computational predicted functional interaction data among proteins and genes to extend and annotate the submitted gene list by their interactive biological pathways and visualize its inferred interaction network accordingly. We used GeneMANIA to conduct the interaction network inference analysis to TFs enriched in either the striosome or matrix compartments with the arbitrary cutoff of adjusted p-value less than 0.01 and the absolute log2 fold-change greater than 1.

### Gene regulatory network inference through data curation

A gene regulatory network links TFs to their target genes and represents a map of transcriptional regulation. We used all the TFs and their target gene data curated by ORegAnno (*Lesurf et al., 2016*) to build the network. To simplify the network, we only chose the compartmental differentially expressed TFs that are high on the hierarchy. In other words, only the differentially expressed TFs that served as a regulator of other differentially expressed TFs were chosen as the candidates of a gene regulatory network.

### Gene set enrichment analysis

GO (*Subramanian et al., 2005*) was performed using ranked list of differential gene expression with parameters set to 2000 gene-set permutations and gene-set size between 15 and 200. The gene-sets included for the GSEA analyses were obtained from Gene Ontology (GO) database (GOBP_All-Pathways), updated September 01, 2019 (http://download.baderlab.org/EM_ Genesets/). An enrichment map (version 3.2.1 of Enrichment Map software *Merico et al., 2010*) was generated using Cytoscape 3.7.2 using significantly enriched gene-sets with an FDR <0.05. Similarity between gene-sets was filtered by Jaccard plus overlap combined coefficient (0.375). The resulting enrichment map was further annotated using the AutoAnnotate Cytoscape App.

### Data processing

The preprocessing of ATAC-seq data involved the following steps:

#### Alignment

Sequencing reads were provided by the sequencing center demuxed and with adaptors trimmed. Reads from each sample were aligned on GRCh38-mm10 reference genome using the STAR aligner (*Dobin et al., 2013*) (v2.5.0) with the following parameters:

```
alignIntronMax 1
outFilterMismatchNmax 100
alignEndsType EndToEnd
outFilterScoreMinOverLread 0.3
outFilterMatchNminOverLread 0.3
```

This produced a coordinate-sorted BAM file of mapped paired-end reads for each sample. We excluded reads that: (1) mapped to more than one locus using SAMtools (*Li et al., 2009*), (2) were duplicated using PICARD (v2.2.4; http://broadinstitute.github.io/picard; *Picard, 2016*), and (3) mapped to the mitochondrial genome.

#### Quality-control (QC) metrics

The following quality-control metrics were calculated for each sample: (1) total number of initial reads, (2) number of uniquely mapped reads, (3) fraction of reads that were uniquely mapped and additional metrics from the STAR aligner, (4) Picard duplication and insert metrics, and (5) normalized strand cross- correlation coefficient (NSC) and relative strand cross-correlation coefficient (RSC), which are metrics that use cross-correlation of stranded read density profiles to measure enrichment independently of peak calling. *Figure 4—source data 1* describes the main QC metrics. The bigWig tracks for each sample were manually inspected. None of the libraries failed QC and visual inspection and eight libraries were subjected to further analysis.

#### Selection and further processing of samples meeting quality control

We subsequently subsampled samples to a uniform depth of 10 million paired-end reads and merged the BAM-files of samples from the same cell type. We called peaks using the Model-based Analysis of ChIP-Seq (MACS) (*Zhang et al., 2008*) v2.1 (https://github.com/macs3-project/MACS; *Liu, 2015*). It models the shift size of tags and models local biases in sequencability and mapability through a dynamic Poisson background model. We used the following parameters:

```
—keep-dup all
—shift −100
—extsize 200
```

—nomodel

We created a joint set of peaks requiring each peak to be called in at least one of the merged BAM-files. That is, if a peak was identified in just one or more samples it was included in the consensus set of peaks. If two or more peaks partially overlapped, the consensus peak was the union of bases covered by the partially overlapping peaks. After removing peaks overlapping the blacklisted genomic regions, 69,229 peaks remained. We subsequently quantified read counts of all the individual non-merged samples within these peaks, again, using the feature counts function in RSubread (*Liao et al., 2014*) (v.1.15.0). We counted fragments (defined from paired-end reads), instead of individual reads, that overlapped with the final consensus set of peaks. This resulted in a sample by peak matrix of read counts, obtained using the following parameters:

```
allowMultiOverlap = F,
isPairedEnd = T,
strandSpecific = 0,
requireBothEndsMapped = F,
minFragLength = 0,
maxFragLength = 2000,
checkFragLength = T,
countMultiMappingReads = F,
countChimericFragments = F
```

## Differential analysis of chromatin accessibility

To identify genomic regions with significant regional differences in chromatin structure among the two cell types, we performed a statistical analysis of chromatin accessibility. Here, chromatin accessibility was assessed by how many ATAC-seq reads overlap a given OCR: the higher the read count, the more open the chromatin is at a given OCR. For this, we performed the following steps:

### Read counts

As a starting point, we used the sample-by-OCR read count matrix described in the previous section (eight samples by 69,229 OCRs). From here, we subsequently removed nine OCRs using a filtering of 0.5 CPM in at least 50% of the samples, resulting in our final sample-by-OCR read count matrix (eight samples by 69,220 OCRs). Next, we normalized the read counts using the trimmed mean of M-values (TMM) method (*Robinson et al., 2010*).

### Covariate exploration

To explore factors affecting the observed read counts, we examined several biological and technical sample-level variables. For these covariates (e.g. number of peaks called in the sample, chrM metrics, RSC and NSC, and Picard insert metrics), we normalized to the median of the cell. We next assessed the correlation of all the covariates with the chromatin accessibility values in the normalized read count matrix to determine which of these variables should be candidates for inclusion as covariates in the differential analysis. We did this using a principal component analysis of the normalized read count matrix and by examining which variables were significantly correlated with the high-variance components (explaining > 1% of the variance) of the data. We did not identify any significant association even when we used a lenient false discovery rate threshold of 0.2.

### Differential analysis

We used the edgeR package (*Robinson et al., 2010*) to model the normalized read counts by negative binomial (NB) distributions. The estimateDisp function was used to estimate an abundance-dependent trend for the NB dispersions (*McCarthy et al., 2012*). To normalize for compositional biases, the effective library size for each sample was estimated using the TMM approach as described above. For each open chromatin region, we applied the following model for the effect on chromatin accessibility of each variable on the right-hand side:

chromatin accessibility ~ cell type.

Then, for each OCR, the cell type coefficient was statistically tested for being non- vanishing. A quasi-likelihood (QL) F-test was conducted for each OCR using the glmQLFTest function

(*Lund et al., 2012*) from the edgeR package, with robust estimation of the prior degrees of freedom. p-values were then adjusted for multiple hypothesis testing using false discovery rate (FDR) estimation, and the differentially accessible regions of chromatin were determined as those with an estimated FDR below, or at, 5%.

### Annotation of OCRs

We used the gene annotations form the org.Mm.eg.db (version 3.8.2) package for all analyses in this paper. We assigned the closest gene and the genomic context of an ATAC-seq OCR using ChIP-Seeker (*Dixon et al., 2015*). The genomic context was defined as promoter (+/- 3 Kb of any TSS), 5'-UTR, 3'-UTR, exon, intron, distal intergenic and downstream. TF binding motif analysis of ATAC-seq data was performed using HOMER suit function findMotifsGenome.pl tool. Differential footprinting analysis was performed using TOBIAS (https://doi.org/10.1101/869560) using TF motifs from Jaspar database (doi: 10.1093/nar/gkx1126).

## Acknowledgements

This research was supported by the National Institute of Neurological Disorders and Stroke Grant R01-NS-100529 and the Collaborative Center for X-linked Dystonia Parkinsonism (to LME and MEE). The Taube HD Stem Cell Consortium also provided support (LME) as well as NLM fellowship grant T15LM007442 (HB). A postdoctoral fellowship was provided to KTT from the Collaborative Center for X-linked Dystonia Parkinsonism.

## Additional information

### Funding

| Funder | Grant reference number | Author |
| --- | --- | --- |
| National Institute of Neurological Disorders and Stroke | R01-NS-100529 | Maria-Daniela Cirnaru<br>Sicheng Song<br>Kizito-Tshitoko Tshilenge<br>Sean D Mooney<br>Lisa M Ellerby<br>Michelle E Ehrlich |
| Collaborative Center for X--Linked Dystonia Parkinsonism at Massachusetts General Hospital | | Kizito-Tshitoko Tshilenge<br>Lisa M Ellerby<br>Michelle E Ehrlich |
| NLM | T15LM007442 | Houda Benlhabib |
| The Taube Family Program in Regenerative Medicine Genome Editing for Huntington's Disease at Buck Institute for Research on Aging | | Lisa M Ellerby |

The funders had no role in study design, data collection and interpretation, or the decision to submit the work for publication.

### Author contributions

Maria-Daniela Cirnaru, Conceptualization, Data curation, Formal analysis, Supervision, Validation, Investigation, Visualization, Methodology, Writing - original draft; Sicheng Song, Data curation, Software, Formal analysis, Validation, Investigation, Visualization, Methodology; Kizito-Tshitoko Tshilenge, Data curation, Formal analysis, Supervision, Validation, Investigation, Methodology; Chuhyon Corwin, Formal analysis, Investigation; Justyna Mleczko, Validation, Investigation; Carlos Galicia Aguirre, John Fullard, Software, Formal analysis, Validation, Investigation, Visualization, Methodology; Houda Benlhabib, Supervision, Validation, Investigation, Methodology; Jaroslav Bendl, Software, Formal analysis, Validation, Investigation, Methodology; Pasha Apontes, Data curation, Software, Formal analysis, Validation, Visualization; Jordi Creus-Muncunill, Formal analysis, Validation, Investigation, Visualization; Azadeh Reyahi, Ali M Nik, Peter Carlsson, Resources; Panos

Roussos, Data curation, Software, Formal analysis, Supervision, Validation, Investigation, Visualization, Project administration; Sean D Mooney, Data curation, Software, Formal analysis, Supervision, Validation, Investigation, Methodology; Lisa M Ellerby, Conceptualization, Resources, Data curation, Software, Formal analysis, Supervision, Funding acquisition, Validation, Investigation, Visualization, Methodology, Writing - original draft, Project administration, Writing - review and editing; Michelle E Ehrlich, Conceptualization, Formal analysis, Supervision, Funding acquisition, Investigation, Writing - original draft, Writing - review and editing

### Author ORCIDs
Houda Benlhabib (iD) http://orcid.org/0000-0002-7776-1508
Lisa M Ellerby (iD) https://orcid.org/0000-0002-9050-7977
Michelle E Ehrlich (iD) https://orcid.org/0000-0001-9397-686X

### Ethics
Animal experimentation: Animal procedures were conducted in accordance with the NIH Guidelines for the Care and Use of Experimental Animals and were approved by the Institutional Animal Care and Use Committee of our institutions (LA09-00272, 16-0847 PRYR1).

### Decision letter and Author response
Decision letter https://doi.org/10.7554/eLife.65979.sa1
Author response https://doi.org/10.7554/eLife.65979.sa2

## Additional files
### Supplementary files
- Source data 1. Differential expression bioinformatics.
- Source data 2. ATAC-seq samples.
- Source data 3. ATAC-seq results.
- Source data 4. Motif analysis.
- Transparent reporting form

### Data availability
All the raw and normalized count are in the GEO data depository (accession codes: GEO is GSE143727 for ATAC-seq and GSE143276 for the RNAseq). Figures 2 and 5-9 have associated raw data. All the data will be available a year from the date of the manuscript publication. The analysis was done using free, publicly available software programs and libraries cited within the methods.

The following datasets were generated:

| Author(s) | Year | Dataset title | Dataset URL | Database and Identifier |
|---|---|---|---|---|
| Cirnaru M, Song S, Tshilenge K, Corwin C, Mleczko J, Aguirre CG, Benlhabib H, Bendl J, Apontes P, Fullard JF, Creus-Muncuni J, Reyah A, Nik AM, Carlsson P, Roussos P, Mooney SD, Ellerby LM, Ehrlich ME | 2020 | Unbiased identification of novel transcription factors in striatal compartmentation and striosome maturation | https://www.ncbi.nlm.nih.gov/geo/query/acc.cgi?acc=GSE143727 | NCBI Gene Expression Omnibus, GSE143727 |
| Cirnaru M, Song S, Tshilenge K, Corwin C, Mleczko J, Aguirre CG, | 2020 | Unbiased identification of novel transcription factors in striatal compartmentation and striosome maturation | https://www.ncbi.nlm.nih.gov/geo/query/acc.cgi?acc=GSE143276 | NCBI Gene Expression Omnibus, GSE143276 |

Benlhabib H, Bendl J, Apontes P, Fullard JF, Creus-Muncuni J, Reyah A, Nik AM, Carlsson P, Roussos P, Mooney SD, Ellerby LM, Ehrlich ME

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
