## [Decision Letter]

**Acceptance summary:**

This paper examines the two compartments of the striatum – the striosome and the matrix. Through transcriptomic and chromatin profiling they identify that Foxf2 and Olig2 are essential for striosomal compartment identity, perhaps regulated by Stat1. The striatum is implicated in many neurodegenerative diseases and addictive behaviours, so the findings are potentially of broad interest.

**Decision letter after peer review:**

Thank you for submitting your article "Unbiased identification of novel transcription factors in striatal compartmentation and striosome maturation" for consideration by *eLife*. Your article has been reviewed by 3 peer reviewers, including Sonia Sen as the Reviewing Editor and Reviewer #2, and the evaluation has been overseen by Marianne Bronner as the Senior Editor.

The reviewers have discussed their reviews with one another, and the Reviewing Editor has drafted this to help you prepare a revised re-submission. The recommendations include new experiments and should you choose to do them we would be pleased to consider the manuscript as a new submission and approach the same reviewers.

Summary:

This paper sets out to understand the transcription factors (TFs) that determine how cells in the striatum differentiate into their principal substructures, and uses that information to derive these specific cell types from human stem cells. The striatum is implicated in many neurodegenerative diseases and addictive behaviours, so the findings are potentially of broad interest. However, more data are required to conclusively demonstrate the functional importance of the identified TFs.

Essential revisions:

While the data presented in this manuscript are important and relevant, we found them to be insufficient to support the claims made: that the authors had identified a sriosomal-specific TF network. The strongest evidence supporting the authors' claim is that in the absence of Foxf2 the striosomes do not mature well. We list here three new experiments that we think are essential to support the claims made. These are:

1. An in vivo overexpression analysis of Foxf2 to demonstrate whether it can drive striosomal compartmentalisation.

2. An in vivo demonstration that downregulation of Olig2 affects striosomal maturation/compartmentalisation.

3. An in vivo demonstration that overexpression of Olig2 affects striosomal maturation/compartmentalisation.

In addition to these experiments we have recommendations for better representations of the following:

1. The iPSC data: The authors suggest that Stat1 is upstream of Foxf2 and Olig2 and use this information to differentiate iPSCs into striosomal identity. Their manipulations fall short of doing this, and instead they end by proposing that Olig2 alone, followed by Olig2/Stat1 in combination, would represent a protocol for induction of a striosome fate. Other methods exist to differentiate iPSCs to striatal identity. So, while they claim their protocol is superior to existing methods, the authors should provide information about the markers used and how they compare quantitatively and qualitatively to other methods.

2. The quality of the microscopy and image representation: The authors need to revisit their microscopy images. It will be helpful to have annotations to guide the reader (including demarcations indicating striosome vs matrix, arrows guiding the reader's attention, and clearly defined high magnification insets), using greyscale to represent single channel images, using the same magnification for the single channels and the merge, among others. In some images, such as 6E it's hard to tell background from the staining (is it background or does OCR drive Olig2 expression in the matrix region as well?). They might consider referring to this preprint that highlights common pitfall and also offers solutions:

[https://www.biorxiv.org/content/10.1101/2020.10.08.327718v1].

*Reviewer #1:*

This study by Cirnaru and colleauges aims to identify new transcription factors and open chromatin regions that are important for the differentiation of striatal neurons in the striosomal compartment. Authors took advantage of the Nr4a1-eGFP mouse to isolate striosome cells and compared them to matrix ones (GFP neg). Transcriptomic analysis followed by bioinformatics and functional analyses reveal a list of transcription factors that are enriched in striosomes. They identify STAT as a factor that might be important for the development of the striosome compartments via Foxf2 and Olig2. They test this via expression studies, mutant analysis and misexpression studies for Foxf2 and expression analysis and mixepression studies for the other two genes. Finally, they use this gene network in various combinations to drive compartmentation in vitro using human iPSCs. This study is of interest as the molecular identity, (even function) of striosome and matrix compartments and how cells compartmentalize into these two areas is poorly understood.

Major strengths:

Authors provide a detailed transcriptomic analysis of transcription factors (TFs) that might participate in the compartmentalization of striatal cells into striosomes. These data are likely to be important to neuroscientists who work on striatal development and on informing protocols aimed at differentiating iPSCs into specific striatal subtypes.

Weaknesses:

While the authors identify several TFs and validate some of them to some extent, the study remains limited as they do not thoroughly address whether the factors they identify are involved in specifying striosomal identity and compartmentation in vivo. Also, it is unclear how these TFs interact functionally and in a spatiotemporal manner to induce striosome compartmentalization.

*Reviewer #2:*

This manuscript takes on a developmental examination of the two compartments of the striatum – the striosome and the matrix – both of which are populated by the well characterised medium spiny neurons of the d and i types. To this end, they FACS sort cells of these compartments and perform bulk transcriptomics and open chromatin profiling. They focus on differentially expressed transcription factors and using network analyses, identify STAT as a factor that might be important for the development of the striosome compartments via Foxf2 and Olig2. They test this via expression studies, mutant analysis and misexpression studies for Fox2 and expression analysis and mixepression studies for the other two genes. Finally, they use this gene network in various combinations to drive compartmentation in vitro using human iPSCs.

*Reviewer #3:*

Cirnaru et al., used a transcriptomics based approach to identify transcription factors (TFs) crucial for striatal compartmentation and maturation of the striosomes.

Nr4a1 is expressed in the medium spiny projection neurons (MSNs) of the striosomes. They used Nr4a1- EGFP mouse and dissected out postnatal day (PND) 3 striatum to separate striosomes from the matrix neurons by FACS. The segregation was verified with a qPCR analysis of the enrichment of marker gene expression namely Nr4a1, Ppp1r1b and Oprm1. RNA-seq was then performed on the EGFP+ and EGFP- cells to identify differentially expressed genes and TFs. TFs namely Irx1, Irx2, Foxf2 and Olig2 were narrowed down for further validation. Irx1 colocalised with a subset of DARPP-32 striosomes. They analysed the RNA-seq data bioinformatically to generate TF co-expression networks for striosomes and matrix regions. Further, they analysed bio-informatically co-regulated target genes of Foxf2 and Olig2.

Next, they performed ATAC-seq on the two segregated cell population and determined differential TF binding motifs being selectively enriched in the two populations.

They then show that Foxf2 protein and RNA colocalises with DARPP-32 in the striosomes and that in Foxf2-null mice the striosomes marked by focal staining of DARPP-32 were absent and instead a diffused staining of DARPP-32 was seen. This suggests that Foxf2 is required for striosome compartmentation and maturation.

Overexpression of Foxf2 in mouse striatal neuronal cultures increases DARPP-32 staining and upregulation of both markers of striosome and matrix regions. This is supportive of a role for Foxf2 in the overall specification of striatal neurons and not a striosome specific role.

Further, Olig2 protein and mRNA colocalised with DARPP-32 in striosomes though Olig2+ve cells are visible throughout the striatum. Overexpression of Olig2 in striatal neuronal cultures increased the expression of striosome markers Oprm1, Foxp2 and Rasgrp1 but not matrix marker Calb1 suggesting a role for Olig2 in striosome maturation.

The OCR – 4.4kb downstream peak to Olig2 gene seen only in EGFP+ striosomes was cloned upstream of m-cherry and analysed for its ability to drive the expression of mcherry exclusively in the striosomes.

Next, the authors move onto STATs and identify both Stat1 and 3 to be enriched in EGFP+ cells. Again overexpression of STAT1 led to increased DARPP-32+ cells and upregulation of both markers of striosome and matrix regions.

Next, the authors compare their RNA-seq data to published Huntington's disease (HD) differentially regulated RNA-seq datasets from both human and mouse and find significant overlap between matrix and striosome compartment to HD gene expression changes.

Finally, they overexpress Foxf2, Olig2 and STATs alone or in combinations to generate striatal MSNs from human iPSCs.

The manuscript presents an interesting analysis of differentially expressed TFs some of which are novel, in the striosome vs matrix compartments of the striatum but the data presented do not show conclusively the precise functional role, if any, for any of these TFs.

Major comments on the manuscript:

1) Figure 2A-F- The use of Nr4a1- EGFP indeed sorted the striosome vs matrix regions in the striatum led to the identification of differentially expressed TFs in the two regions namely Irx1, Irx2, Foxf2 and Olig2.

2) Figure 5C- In the absence of Foxf2 the striosomes do not mature well and diffuse staining of diffused staining of DARPP-32 was seen. This is the only piece of evidence supporting the claim by the authors that their study has indeed identified region-specific TFs with a role in striosome compartmentation and maturation. The rest of the data are at best correlative as they only show localisation of the TFs with DARPP-32- a marker for striosomes.

3) Figure 6- In the absence of Olig2 loss of function experiments it is difficult to determine its precise role in striosome maturation.

4) Figure 6E- The mcherry was seen colocalising with DARPP-32 marker for striosomes but is also sparsely seen in the matrix region. Is this background or that the OCR drives Olig2 expression in the matrix region as well albeit at lower levels as compared to its expression in striosomes.

5) Figure 8A- It is not clear what the authors are hypothesising here given that the gene expression from both matrix and striosome compartments show significant overlap with HD gene expression changes and that all the datasets are from the striatum either from mouse or human and the only difference is whether they are from control or patients. It is no surprise then that both matrix and striosome compartments do show changes as would if the entire striatum was to be taken.

6) Overall – Though the approach is unique and TFs identified are potentially novel regulators of striosome compartmentation and maturation, the data presented in the manuscript currently is not convincing enough corroborating the claims by the authors.

---

## [Author Response]

Essential revisions:While the data presented in this manuscript are important and relevant, we found them to be insufficient to support the claims made: that the authors had identified a sriosomal-specific TF network. The strongest evidence supporting the authors' claim is that in the absence of Foxf2 the striosomes do not mature well. We list here three new experiments that we think are essential to support the claims made. These are:1. An in vivo overexpression analysis of Foxf2 to demonstrate whether it can drive striosomal compartmentalisation.

For this purpose, we utilized a previously characterized FOXF2 BAC transgenic line (Moussavi Nik et al., 2013) overexpressing human FOXF2. At postnatal day 3 (PND3), the area occupied by striosomes as detected by DARPP-32 immunostaining was increased by over 40% (new Figure 5E,F). In addition, multiple markers of striosome maturation are up-regulated (new Figure 5G), confirming that Foxf2 can drive striosomal compartmentalization.

2. An in vivo demonstration that downregulation of Olig2 affects striosomal maturation/compartmentalisation.

For this purpose, we utilized a mouse in which Cre recombinase had been inserted into the Olig2 gene, resulting in absence of functional protein expression (JAX Stock No. 025567), so these mice were analyzed in the homozygote state. They were viable until late on the first day of life, and therefore were collected early on the first day of life. Striosomes are noticeably underdeveloped and different from Foxf2-null striatum, there was almost no detectable DARPP-32 (Figure 6E). Markers of striosome were reduced while markers of the matrix were unchanged (Figure 6F).

3. An in vivo demonstration that overexpression of Olig2 affects striosomal maturation/compartmentalisation.

Our attempts at in utero overexpression by viral injection were unsuccessful, so we performed intracerebroventricular injection of AAV-Olig2 on the day of birth. Although not ideal as obviously striosomes are already present at the time of injection, and overexpression is not immediate, we did observe a surprising decrease in some of the striosome markers and an increase in at least one of them, indicating a definite effect of postnatal overexpression of Olig2 on striosomes, but one that is not straightforward mechanistically and requires further investigation during early development (Supplementary Figure 4E,F).

We have also added references to the TsDN65 (“Down’s mouse) literature, as Olig2 is one of the genes in the triplicated region and is over-expressed. Reports indicate an increase in neurogenesis and production of GABAergic interneurons with direct interactions with the TFs Dlx1 and Lhx8 (Liu et al., 2015; Xu et al. 2019). Although we were unable to obtain these mice during this period, we will of course be interested in analyzing the striatum in this line.

In addition to these experiments we have recommendations for better representations of the following:1. The iPSC data: The authors suggest that Stat1 is upstream of Foxf2 and Olig2 and use this information to differentiate iPSCs into striosomal identity. Their manipulations fall short of doing this, and instead they end by proposing that Olig2 alone, followed by Olig2/Stat1 in combination, would represent a protocol for induction of a striosome fate. Other methods exist to differentiate iPSCs to striatal identity. So, while they claim their protocol is superior to existing methods, the authors should provide information about the markers used and how they compare quantitatively and qualitatively to other methods.

Pertinent to the mouse and primary culture experiments, as well as the iPSCs, we have added Table 1 to list the markers which we used, emphasizing that for some of them, distribution changes between PND3 and adult.

There are several reasons our methods improve differentiation of NSCs into MSNs and

striosomal identity. First the methods to make MSNs from NSCs requires weeks to months to differentiate the cells into cells with MSN makers. Our induction takes 4 days and the DARPP- 32 and MOR markers are robustly expressed along with other striosomal markers which are quantified by RT-PCR in Figure 8B. We found that both Olig2 and Foxf2 are are required to robustly express multiple markers of striosomal identity including DARPP-32 and MOR. Finally, although not part of this report, we are working towards an striatal organoid model and these TFs are key to making striosomal MSNs in this system. This will be the subject of a separate publication.

2. The quality of the microscopy and image representation: The authors need to revisit their microscopy images. It will be helpful to have annotations to guide the reader (including demarcations indicating striosome vs matrix, arrows guiding the reader's attention, and clearly defined high magnification insets), using greyscale to represent single channel images, using the same magnification for the single channels and the merge, among others. In some images, such as 6E it's hard to tell background from the staining (is it background or does OCR drive Olig2 expression in the matrix region as well?). They might consider referring to this preprint that highlights common pitfall and also offers solutions: [https://www.biorxiv.org/content/10.1101/2020.10.08.327718v1].

The microscopy has been revised and hopefully improved taking into account this very useful advice. The color has been changed and striosomal demarcation and arrows have been added. The red/green combination is not used, but rather magenta/green is the combination which is utilized.

Striosomes and particular cell types are now highlighted with arrowheads throughout the figures.

Regarding the Olig2 OCR vector: Please note that it drives expression to striosome neurons and in the matrix, to individual cells that are DARPP-32-negative, and likely represent cells of the oligodendroglial lineage. We apologize for not previously highlighting this finding.